# Event Coreference Data (Almost) for Free: Mining Hyperlinks from Online News

**Michael Bugert** and **Iryna Gurevych**

*Ubiquitous Knowledge Processing Lab (UKP)*
*Department of Computer Science, Technical University of Darmstadt*
*https://www.ukp.tu-darmstadt.de*

## Abstract

Cross-document event coreference resolution (CDCR) is the task of identifying which event mentions refer to the same events throughout a collection of documents. Annotating CDCR data is an arduous and expensive process, explaining why existing corpora are small and lack domain coverage. To overcome this bottleneck, we automatically extract event coreference data from hyperlinks in online news: When referring to a significant real-world event, writers often add a hyperlink to another article covering this event. We demonstrate that collecting hyperlinks which point to the same article(s) produces extensive and high-quality CDCR data and create a corpus of 2M documents and 2.7M silver-standard event mentions called *HyperCoref*. We evaluate a state-of-the-art system on three CDCR corpora and find that models trained on small subsets of *HyperCoref* are highly competitive, with performance similar to models trained on gold-standard data. With our work, we free CDCR research from depending on costly human-annotated training data and open up possibilities for research beyond English CDCR, as our data extraction approach can be easily adapted to other languages.[1]

## 1. Introduction

Cross-document event coreference resolution (CDCR) is the task of identifying and clustering mentions of real-world events in a given collection of documents. For example, CDCR systems need to decide whether the sentences "On Monday, Lindsay Lohan checked into rehab in Malibu" and "Ms. Lohan entered a rehab facility" from two different documents refer to the same event, by taking temporal, spatial and other cues from the document contexts into account. CDCR is a vital preprocessing step for downstream multi-document tasks such as question answering or fact checking.

Annotation of CDCR is laborious and expensive, requiring expert annotation which can span weeks for several hundred documents [Cybulska and Vossen, 2014b, Vossen et al., 2018]. Crowdsourcing annotation has been proposed, however this requires extensive training of annotators [Bornstein et al., 2020] or post-processing by experts [Bugert et al., 2020, 2021], precluding large-scale studies. Annotating CDCR data in a different language requires great effort since language-specific guidelines [Minard et al., 2016] and enough annotators with proficiency in the target language are required. As a consequence, CDCR corpora had to compromise on size, domain coverage, and the density of annotated mentions and coreference links, as well as language coverage.

---

1. Data and model available at https://github.com/UKPLab/emnlp2021-hypercoref-cdcr

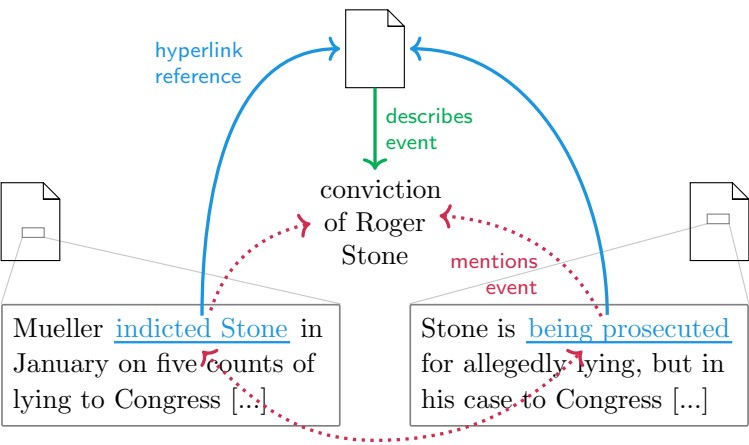

Figure 1: Three online news articles interconnected by two hyperlinks. The relation between the underlined hyperlink anchors can be interpreted as cross-document event coreference.

This data bottleneck is problematic for three reasons. Firstly, recent state-of-the-art CDCR systems are based on pretrained language models [Devlin et al., 2019, Liu et al., 2019] fine-tuned with supervised learning on human-annotated corpora [Zeng et al., 2020, Yu et al., 2020, Caciularu et al., 2021, Cattan et al., 2021]. Achieving top results with such models still requires high-quality labeled data for fine-tuning [Gururangan et al., 2020], yet the size of current corpora is insufficient for training large systems. Secondly, because corpora need to make compromises on domain coverage, the domain coverage of all existing CDCR corpora is limited even when combined. This holds back research on open-domain CDCR systems which could increase the (currently limited) applicability of CDCR in downstream tasks [Bugert et al., 2021]. Thirdly, because test splits consist only of a few hundred documents (far less than what downstream applications may require), scalability to large corpora is a problem which could not be tackled so far [Bugert et al., 2021].

To overcome this gap, we leverage hyperlinks in online news articles: When referencing a significant real-world event in the body of an article, writers often add a hyperlink to a different article covering this event. We conjecture that by collecting hyperlinks which point to the same article(s) and interpreting anchor texts as mention spans, high-quality cross-document event coreference links can be retrieved quickly and in large quantity (see Figure 1). To this end, we devise a data extraction pipeline which mines such datasets automatically from Common Crawl[2] and apply it to create the HYPERCOREF corpus, consisting of 40 news outlets with over 2M mentions in total, far exceeding the size of existing CDCR corpora. HYPERCOREF achieves broader coverage in event types compared to manually annotated corpora. In experiments with a state-of-the-art CDCR model [Cattan et al., 2021], we evaluate the relation between the amount of gold training data and test performance across three CDCR corpora: ECB+, FCC-T, and GVC. We make the remarkable observation that models trained entirely on silver-standard data from HYPERCOREF perform on a similar level as models trained on gold-standard data (between 4 pp. CoNLL F1 worse and 4 pp.

---

2. https://commoncrawl.org/

better, depending on the corpus at hand). Overall, our findings lift the dependency on gold data for training CDCR systems and pave the way for large, robust and potentially multilingual systems, as our data extraction approach can be easily adapted to any language found on the web. Our contributions are:

**C1.** A novel approach for acquiring silver-standard cross-document event coreference links from hyperlinks,

**C2.** HYPERCOREF, a large corpus created with this approach, and its analysis compared to gold-standard CDCR corpora,

**C3.** out-of-domain transfer experiments with a state-of-the-art CDCR system, certifying the quality of this data.

## 2. Fundamentals

We define cross-document event coreference resolution (CDCR) and its relation to hyperlinks in news.

**Task Definition**   CDCR consists of two steps: (1) identifying mentions of real-world or hypothetical events in a collection of documents (**event mention detection**), and (2) recognizing which of these mentions refer to the same events, thereby producing a cross-document clustering of mentions (**coreference resolution**). Event mentions are commonly defined by four components: their action (*checked into*), participants (*Lindsay Lohan*), time (*on Monday*) and location (*rehab in Malibu*) [Cybulska and Vossen, 2014a]. Actions are the centerpiece of event mentions, and their token span is the main representative of an event mention in text. We also refer to this span as a **mention span**.

**Hyperlinks in News**   To establish the context of a recent news development, news journalists make reference to other events which have caused, influenced or are otherwise related to the recent newsworthy event. In online news, such references are often marked with a hyperlink to another article which covers the referenced event in greater detail. These hyperlinks can (with some margin of error) be interpreted as cross-document coreference links: The hyperlink's **anchor text** (its clickable text region) corresponds to an event mention's action, and the target URL identifies the referenced event. A pair of hyperlinks which point to the same URL but are located in different articles then correspond to two event mentions connected by a CDCR link. This is exemplified in Figure 1.

We propose to collect CDCR data by mining hyperlinks from online news. In the next section, we explain our data pipeline creating such data and the key issues one needs to overcome in the process.

## 3. Data Extraction

Following an explanation of our data extraction pipeline, we describe its application for creating the HYPERCOREF corpus which we then compare to expert-annotated CDCR corpora.

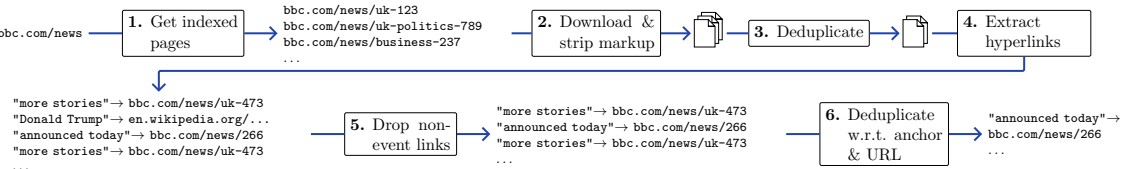

Figure 2: Cross-document event coreference data pipeline relying on Common Crawl.

## 3.1 Pipeline

We apply our pipeline on one news outlet at a time for greater computational efficiency. The following steps are visualized in Figure 2. Given a news outlet:

1. We source documents from Common Crawl (CC), a public repository of crawled web pages. We retrieve a list of all pages available for this news outlet from the CC index.
2. We download these web pages, remove excess markup and detect publication dates with the `newspaper3k` framework[3].
3. We drop pages without in- or outgoing hyperlinks. A large portion of pages tend to be duplicates. We deduplicate pages based on their textual content using locality sensitive hashing [Leskovec et al., 2020, ch. 3] and clustering of pages by their TF-IDF cosine similarity in a two-stage approach.
4. We extract hyperlinks, remove their URL query strings, and harmonize the boundaries of their anchor text (trimming of whitespace, exclusion of punctuation).
5. The removal of hyperlinks which refer to non-eventive content such as overview pages, product reviews, or affiliate products plays a key role. We apply a series of filtering steps:
   (a) Links whose target domain (`amazon.com`, `facebook.com`, etc.) mismatches the article's domain are removed.
   (b) We make the assumption that the majority of links do refer to eventive content, and that their URLs share syntactic similarities. We build a prefix tree from all URLs and retain only those links whose URLs are part of the 90% most frequent prefixes.
   (c) Links to pages with high indegree or from pages with high outdegree are removed.
   (d) A set of handwritten rules targeting URLs and anchor texts is applied, which for example remove URLs containing `/tag/`, `/category/`, or anchor texts such as "*click here*".
6. Groups of hyperlinks sharing the same anchor text and target URL are removed entirely. This eliminates any remaining links to hub pages or "read more"-type links appearing out of context on multiple pages.

All pipeline steps except for the handwritten rules in step 5d) are language-independent, hence the pipeline can be easily applied to news in any language found on the web. We limit ourselves to English news in this work since the majority of gold-standard CDCR corpora are in English. In several test runs with Dutch, German, and French news we observed results of similar quality to those created from English news.

---

3. https://pypi.org/project/newspaper3k/

| Corpus | Docs | Mentions* | Clusters* | Uniq. verbs |
|---|---|---|---|---|
| ECB+ | 0.98 k | 4.81 k | 0.72 k | 573 |
| FCC-T | 0.45 k | 3.18 k | 0.26 k | 91 |
| GVC | 0.51 k | 6.93 k | 1.05 k | 71 |
| HYPERCOREF | 2.05 M | 2.74 M | 0.81 M | 4071 |
| ABC | 38.40 k | 53.37 k | 16.20 k | 1807 |
| BBC | 98.16 k | 142.14 k | 37.58 k | 1520 |
| *+38 others* | 1.92 M | 2.55 M | 0.76 M | 4071 |

| ECB+ | HYPERCOREF | |
|---|---|---|
| | ABC | BBC |
| 1. state.v.01 | state.v.01 | win.v.01 |
| 2. kill.v.01 | announce.v.01 | defeat.n.01 |
| 3. die.v.01 | kill.v.01 | beat.v.01 |
| 4. earthquake.n.01 | make.v.01 | join.v.01 |
| 5. announce.v.01 | report.v.01 | lose.v.01 |
| 6. collar.v.01 | die.v.01 | victory.n.01 |
| 7. murder.n.01 | find.v.01 | draw.n.01 |
| 8. death.n.01 | name.v.01 | sign.v.01 |

(a) Size differences. *Excl. singletons.  (b) Top word senses of event mentions

Table 1: Comparison between gold CDCR corpora and the new HYPERCOREF corpus.

## 3.2 The HyperCoref Corpus

We apply our pipeline to 40 news outlets from English-speaking countries to produce HYPERCOREF, a corpus consisting of 2M documents, 0.8M hyperlink clusters ($\hat{=}$ events) and 2.7M hyperlinks ($\hat{=}$ event mentions). The most recent documents included stem from the October 2020 crawl of CC. We compare HYPERCOREF to three CDCR corpora: **EventCorefBank+ (ECB+)** [Cybulska and Vossen, 2014b], a corpus containing news articles from a broad selection of 45 topics, the **Football Coreference Corpus (FCC-T)** [Bugert et al., 2020, 2021] which annotated football match events in sports news and the **Gun Violence Corpus (GVC)** [Vossen et al., 2018] which annotates fine-grained gun violence events in news. A size comparison is shown in Table 1a, demonstrating that HYPERCOREF is several orders of magnitude larger than expert-annotated corpora. Additional analysis (see Appendix A.1) reveals that the distribution of cluster sizes is comparable between HYPERCOREF and expert-annotated corpora, however anchor texts tend to be longer phrases or even entire sentences, opposed to the minimum span annotations of action triggers pursued for the creation ECB+. To keep further analysis and experiments manageable, the remainder of this work focuses on **ABC News** (`abcnews.go.com`) and **BBC News** (`bbc.com`), two large and reputable news outlets.

**Event Types** We heuristically determine the event types contained in each corpus by performing WSD on the syntactic head of mentions against WordNet [Miller, 1995], choosing the most frequent sense and counting word sense occurrences. Table 1b shows the top word senses of three (sub-)corpora. Compared to ECB+, ABC contains a greater proportion of reporting events (cf. Pustejovsky et al. [2003]) and mentions using light verbs, which are challenging for coreference resolution [Hovy et al., 2013, Choubey and Huang, 2017]. BBC consists of events from the sports domain. We count the number of unique verbal word senses of mentions to estimate the event type coverage per corpus (see Table 1a). HYPERCOREF exhibits considerably broader coverage than previous CDCR corpora.

## 3.3 Qualitative Analysis

We manually analyze a total of 300 hyperlinks from the ABC and BBC subcorpora to gain a better understanding of the retrieved data. 70% of these links are accompanied by a

| Referenced article | Article excerpts with hyperlink reference to the article shown left |
|---|---|
| Israeli Police and the Israeli Securities Authority announced the conclusion of a joint investigation into the conduct of Prime Minister Benjamin Netanyahu on Sunday, recommending indictments for both Netanyahu and his wife Sara. | **A1** Israeli Prime Minister Benjamin Netanyahu spoke with UN Secretary General Antonio Guterres on Wednesday to update him on the details of a military operation that will destroy cross-border tunnels between Israel and Lebanon [...].

**A2** Israel's Prime Minister Benjamin Netanyahu should be indicted for bribery, fraud and breach of trust, the country's attorney general said Thursday.

**A3** Netanyahu is the first Israeli prime minister criminally charged while in office. |
| [2015-06-24] St Helens legend Paul Wellens has announced his immediate retirement from rugby league because of a hip injury. [...] Wellens will take up a coaching position at Langtree Park as part of Keiron Cunningham's staff. | **B1** [2015-06-24] They are huge accolades for a quiet man who dedicated his career to one club, and who has announced his retirement at the age of 35 because of a hip problem.

**B2** [2015-07-01] The 22-year-old Australian has agreed to join Saints as full-backs Shannon McDonnell and Jonny Lomax are out, while Paul Wellens retired last week.

**B3** [2015-12-03] Paul Wellens has taken up a player performance coach role at St Helens after his retirement. He announced he was quitting playing in June because of a hip injury and was told he would take up a coaching position which is now confirmed. |

Table 2: Sample event clusters from HYPERCOREF – an ABC News cluster (top) and a BBC News cluster (bottom).

plausible event mention in the same sentence. The remaining 30% refer to topically similar but unrelated events (see A1 in Table 2) or refer to non-event content such as health guides. We analyze if and where the four event components (see Section 2) are found in sentences containing plausible event mentions. For 66% of these links, anchor texts contain the event action. In the remaining 34% of cases, writers oftentimes marked event participants, times, or locations instead to emphasize these aspects (see A1, B2 in Table 2). Although such hyperlinks contradict the common definition of an event mention, we decided against filtering these out since doing so may also have removed event mentions recognizable exclusively by their participants, time or location.[4] For the subset of links where anchor texts contain the event action, 74% of these links exhibit verbal actions (the remainder being predominantly nominal actions).

Overall, HYPERCOREF qualifies as CDCR data, though with inherently noisy clusters and imperfect mention spans. We evaluate the use of HYPERCOREF for training a CDCR system in the next section.

## 4. Experiments

Annotating gold-standard CDCR corpora is a laborious and expensive process, raising the question to which extent such data can be replaced with cheaper to obtain silver-standard (i.e. automatically generated) data for training CDCR models. We investigate this question

---

4. See Appendix A.2 for examples and deeper discussion.

by evaluating the state-of-the-art CDCR system of Cattan et al. [2021] on three gold corpora (ECB+, FCC-T, GVC) and the ABC News and BBC News subcorpora of the silver HyperCoref corpus. We first describe the aforementioned CDCR system and explain how we prepare each corpus for CDCR experiments, then report results for the coreference resolution task. Experiments for the mention detection task are reported in Appendix A.4.2.

**SOTA System**   We choose Cattan et al. [2021] for our experiments since it is the most recent state-of-the-art approach on ECB+ with available implementation. At test time, this system predicts a similarity score for each possible pair of mentions. Using these similarities, agglomerative clustering is performed (with hyperparameter threshold $\tau$) to produce a cross-document clustering. The similarity of two mentions is obtained by separately vectorizing each mention and their document context with RoBERTa [Liu et al., 2019], then combining these vectors with linear layers to produce a single scalar. The system is trained on gold mention pairs with binary labels (coreferring / not coreferring). Akin to Lee et al. [2017], the system can jointly perform coreference resolution and (event) mention detection.[5]

**Evaluation Scenarios**   We evaluate scarcity of gold-standard data in three scenarios of increasing difficulty: in $S_{gg}$, a model has full access to gold-annotated train and dev splits, with an optional equal amount of silver mentions from HyperCoref used during training. In $S_{sg}$, the dev split remains gold but training data is replaced with silver data. Finally, $S_{ss}$ tests out-of-domain transfer using entirely silver train and dev splits.

**Data Preparation**   For ECB+, we use the official splits and filtered sentences specified in the corpus documentation. For FCC-T and GVC, we use the splits of Bugert et al. [2021].

For HyperCoref, we first discard all documents which closely resemble documents from the dev or test splits in either of the three gold-standard corpora so as to guarantee unbiased evaluation later on.[6] We only use clusters consisting of 2 to 10 mentions to strike a balance between cluster sizes and large enough variance in events. To conform a given hyperlink anchor text span to the minimum span annotation of gold-standard corpora (see Section 3.3), we dependency parse the surrounding sentences with CoreNLP [Manning et al., 2014] and choose the syntactic head of the anchor text (including any tokens connected with `compound` or `flat` relations) as the mention span. To keep the training times of the Cattan et al. [2021] system manageable, we limit HyperCoref training data to 25k event mentions in the $S_{ss}$ and $S_{sg}$ scenarios. For $S_{ss}$, we use development splits consisting of 1.7k mentions (ABC) and 4.2k mentions (BBC) which corresponds to 5% of all available mentions for these corpora.

**Results**   We evaluate event coreference resolution performance in-domain on each CDCR corpus, as well as across corpora to measure out-of-domain robustness. Achieving comparable coreference resolution results between the ECB+, FCC-T and GVC corpora requires using gold event mention spans due to non-exhaustive event mention annotations in FCC-T and GVC [Bugert et al., 2021]. We therefore do not use the mention detection mechanism of Cattan et al. [2021] in this set of experiments. We evaluate CDCR with the CoNLL F1

---

5. Please refer to original publication for further details. We report additional training and setup details in Appendix A.3.

6. Using 1,2,3-gram TF-IDF vectors, pages with over 0.25 cosine similarity to gold-standard dev or test subtopics were discarded: 113 documents for ABC, and 89 for BBC.

metric [Pradhan et al., 2012]. Table 3 shows results of the state-of-the-art system for each data scarcity scenario on each corpus. The complete results for all metrics, including the link-based entity-aware coreference metric (LEA) [Moosavi and Strube, 2016], as well as two baseline approaches, are reported in Appendix A.4.

| Scen. | Train | Dev | ECB+ | FCC-T | GVC | H. Mean |
|---|---|---|---|---|---|---|
| $S_{gg}$ | ECB+ | ECB+ | **79.11** | 48.69 | 58.08 | **59.53** |
| | with ABC | ECB+ | 75.18 | 45.82 | **58.26** | 57.37 |
| | with BBC | ECB+ | 75.38 | 48.88 | 57.68 | 58.75 |
| | FCC-T | FCC-T | 61.82 | 47.70 | 49.35 | 52.26 |
| | with ABC | FCC-T | 63.76 | **50.21** | 38.56 | 48.75 |
| | with BBC | FCC-T | 63.24 | 48.17 | 46.49 | 51.65 |
| | GVC | GVC | 66.88 | 44.96 | 45.99 | 50.90 |
| | with ABC | GVC | 64.50 | 46.16 | 51.42 | 52.99 |
| | with BBC | GVC | 67.29 | 45.46 | 46.84 | 51.54 |
| $S_{sg}$ | ABC | ECB+ | **75.42** | 47.40 | 58.13 | 58.19 |
| | | FCC-T | 74.17 | 48.64 | 43.37 | 52.54 |
| | | GVC | 74.55 | 48.50 | 43.69 | 52.70 |
| | BBC | ECB+ | 73.60 | 52.50 | 58.34 | 60.27 |
| | | FCC-T | 71.73 | **53.19** | **59.98** | **60.71** |
| | | GVC | 73.78 | 52.47 | 58.11 | 60.22 |
| $S_{ss}$ | ABC | ABC | **75.04** | 46.82 | 58.75 | 58.02 |
| | BBC | BBC | 66.37 | **51.21** | **60.75** | **58.76** |

Table 3: CDCR performance of Cattan et al. [2021] measured with CoNLL F1. We use gold event mentions, predicted topics, include singletons and score on one meta-document per corpus. Three scenarios are reported: models learned mostly on gold data ($S_{gg}$), mostly on silver data ($S_{sg}$) or entirely on silver data ($S_{sg}$). Results on corpora unseen during optimization are marked in gray. We report the mean of three independent trials.

In $S_{gg}$, the model trained on ECB+ generalizes best. This is due to the broad domain coverage of ECB+ which includes sports and gun violence – the two topics on which FCC-T and GVC specialize. The performance of $S_{gg}$ models trained on an equal amount of gold and silver data from either ABC or BBC is mixed: test performance on individual corpora is at times higher, but aggregated performance across corpora declines. Looking at the most difficult scenario $S_{ss}$, the performance of the models trained and optimized entirely on silver HYPERCOREF data is highly competitive with the in-domain performance of $S_{gg}$ models. The strong results of the BBC model on FCC-T can be attributed to a large portion of football sports news in the BBC subcorpus (see Section 3.2), yet performance on GVC is similarly strong. Performance increases further in the $S_{sg}$ scenario, where gold dev sets are used for early stopping and for choosing the clustering threshold $\tau$.

The most likely explanation as to why $S_{gg}$ models trained on mixed gold and silver data perform worse than $S_{sg}$ and even $S_{ss}$ models appears to be that HYPERCOREF data is most helpful when it is used in large quantities. Using small subsets limits the diversity of training events and bears a greater risk of overfitting to noise.

**Error Analysis** We sample 10 test clusters from each gold-standard corpus and manually compare the predictions of (1) both $S_{ss}$ models, (2) of the $S_{sg}$ models optimized on the respective corpus and (3) of the respective in-domain $S_{gg}$ model without augmentation. On ECB+, we make the common observation [Upadhyay et al., 2016, Barhom et al., 2019, Bugert et al., 2021] that the in-domain model primarily matches event actions with similar surface form. $S_{ss}$ and $S_{gg}$ models are more liberal with merging paraphrases (such as "revealed" or "unveiled") but overmerge unrelated mentions more frequently as a result. Compared to ECB+, FCC-T exhibits greater ambiguity of event actions ("win", "draw", "final" can refer to many different sports matches). The in-domain model rarely clusters such mentions, opposed to the $S_{ss}$ and $S_{sg}$ models which merged such mentions if nearby participant mentions were compatible. Our GVC analysis mirrors these findings, with the BBC $S_{ss}$ and $S_{sg}$ models performing noticeably better merges than other models, particularly for clusters with varied actions ("went off", "shooting", "discharged") where a mention's context matters. In summary, models trained on HYPERCOREF exhibit greater context sensitivity.

## 5. Discussion and Future Work

Expensive data annotation constitutes a bottleneck for research on scalable, open-domain CDCR systems. Addressing this gap, we found that hyperlinks extracted from online news have tremendous potential when used as proxy training data for CDCR: The fact that a model trained on hyperlinks from BBC sports news outperforms in-domain models trained on FCC-T (sports) and GVC (gun violence), and that a model trained on ECB+ performs only 4% pp. CoNLL F1 better than a model trained on hyperlinks from ABC News (see Table 3) demonstrates that hyperlink data can offer both sufficient depth and breadth in domains to enable development of domain-focused and open-domain CDCR models. The proposed data extraction pipeline only requires scraped online news, basic NLP tools[7] and several handwritten filtering rules. Obtaining CDCR training data with sufficient quality is therefore vastly cheaper with this approach than traditional means of annotation, since no annotation guidelines or trained annotators are required. The approach is language-independent save for a set of filtering rules (see Section 3.1). Rules for a particular language can be created by a single individual proficient in that language (or alternatively, by a motivated researcher using machine translation). Hence, we significantly lower the entry hurdle for future CDCR research on languages other than English, for which little to no gold-standard training data exists.

Nevertheless, there are downsides to our proposed approach. By nature, it cannot be applied to text types without hyperlinks such as works of fiction [Sims et al., 2019], e-mail conversations [Dakle et al., 2020] or dialogue [Eisenberg and Sheriff, 2020]. Data scraped from the web is inherently noisy, and while filtering steps can mitigate this issue, noise from imperfect markup cleaning, hyperlinks not referring to eventive content, or different issues remains. Similarly, hyperlink anchor texts oftentimes resemble the mention spans of event mentions as defined in traditional annotation guidelines, but this is not guaranteed

---

7. We required a sentence tokenizer, word tokenizer and dependency parser to conform the data to token-level CDCR format. For use cases in which character-level spans and long anchor texts do not pose an issue, these tools are not required.

(see Section 3.3). Future work may investigate human-in-the-loop annotation [Wang et al., 2021] to resolve such cases on a subset of the data. Training event mention detection systems entirely on hyperlink data is possible (see Appendix A.4.2), however with a deficit in precision and particularly recall since hyperlinks appear less frequently in news data than event mentions do in gold-standard corpora. Doing so can nonetheless be an effective fallback solution for languages for which no gold-standard mention detection corpus is available.

**Future Work** The experiments performed in this work mainly serve the purpose of characterizing the quality of the HYPERCOREF corpus. Future work may explore ways on how to use this data to its full effect to maximize CDCR performance. In particular, the system of Cattan et al. [2021] does not perform document-level inference and does not make use of document publication dates for temporal inference. More advanced CDCR models, such as the recently introduced cross-document language model of Caciularu et al. [2021], may therefore display even greater benefit from HYPERCOREF. Of the scenarios we tested HYPERCOREF in, training a system on HYPERCOREF and optimizing its hyperparameters on gold-standard data produced the best results. Better results may be possible with transfer learning [Pruksachatkun et al., 2020, Vu et al., 2020]. We observed favorable performance when training on only 2 out of 40 news outlets from HYPERCOREF (amounting to just 2% of all mentions available). Future work may exploit the entirety of HYPERCOREF for training large, truly open-domain CDCR models, potentially including additional languages beyond English by adapting our pipeline.

## 6. Related Work

Harvesting NLP datasets from the web has a long history. Sil et al. [2010] extract sequences of verb constructions from webpages to learn common preconditions of actions and events, and Chambers and Jurafsky [2008] extract narrative chains of events from the Gigaword corpus [Graff et al., 2005]. However, these works extract event knowledge from *raw* newswire text, omitting signals from hyperlinks. The *WikiLinks* corpus [Singh et al., 2012] is an entity coreference corpus created by collecting hyperlinks to Wikipedia pages from a web crawl. While large in volume, the dataset does not target events and suffers from low mention ambiguity, since a mention-pair string identity baseline can reach 82% F1. As part of the Wikification task [Roth et al., 2014, Peng et al., 2016], links in the body of Wikipedia articles and their anchors were collected to produce multilingual entity coreference resolution data.

Closest to our approach is Wikipedia Event Coreference (WEC) [Eirew et al., 2021], a recent CDCR corpus created from Wikipedia articles on real-world events and cross-page links. While Wikipedia-based corpora are easier to create than newswire corpora (such as HYPERCOREF) due to standardized markup, their key downside is that encyclopedic text lacks the temporal and spatial anchoring present in newswire (as in "Today, the White house announced" or "Ed Sheeran is coming to town") which considerably lessens their usefulness for event-related tasks. Furthermore, compared to HYPERCOREF, event coverage in WEC is limited to events which the Wikipedia community deemed significant enough to warrant a dedicated article (for each separate language). This excludes high-frequency, high-ambiguity events (resignations, stocks surging, arrests, etc.)[8] which are most challenging to resolve for

---

8. See https://en.wikipedia.org/wiki/Wikipedia:Notability_(events)

CDCR systems and therefore crucial to have as training data. These corpus differences have a direct impact on results, with our broad-coverage ABC $S_{ss}$ model outscoring the identical RoBERTa-based architecture trained by Eirew et al. [2021] on WEC by 5.9% CoNLL F1 when tested on ECB+ (75.04 F1 vs. 69.10 F1).

Recently, Choubey and Huang [2021] investigated automated retrieval of annotations for *within-document* event coreference resolution. Their method is applicable on plaintext news articles, but requires a database of lexical paraphrases for mention identification and a discourse parsing system for filtering. It is unclear whether this method can be successfully applied for CDCR, since the employed discourse-based filtering rules may not be transferable to the cross-document case.

To the best of our knowledge, we are the first to propose a cheap and high-quality data extraction approach specifically for cross-document event extraction and coreference which does not depend on pre-existing resources, combining past work on event extraction from raw newswire text and mining of hyperlinks.

## 7. Conclusion

To overcome the prevalent data bottleneck of the CDCR task, we proposed a new method for cheaply and automatically collecting silver-standard data from hyperlinks in online news. We used this approach to create HyperCoref, a large dataset with over 2M mentions and show that a system trained on a subset of this dataset achieves equivalent performance as the same system trained on expert-annotated corpora. Our data collection approach opens up many avenues for future work, particularly for languages where gold-standard CDCR data is currently scarce or non-existent.

## Acknowledgments

We thank Jan-Christoph Klie, Nafise Sadat Moosavi, Anne-Kathrin Bugert, and the anonymous reviewers for their feedback. This work was supported by the German Research Foundation through the German-Israeli Project Cooperation (DIP, grant DA 1600/1–1 and grant GU 798/17–1), and by the German Research Foundation under grant EC 503/1–1 and GU 798/21–1.

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

## A. Appendix

### A.1 Additional Corpus Statistics

A complete size comparison between the three gold-standard corpora ECB+, FCC-T, and GVC and silver-standard corpora HyperCoref and WEC [Eirew et al., 2021] is shown in Table 4. For HyperCoref, the number of documents shown corresponds to the number of documents containing at least one event mention. The corpus contains an additional 850k documents which only appear as hyperlink targets (i.e., these documents are the seminal documents describing each event cluster) and which are kept for reference or for future experiments.

The distribution of cluster sizes in data retrieved with our pipeline resembles that gold-standard corpora (see Figure 3). Figure 4 demonstrates that hyperlink anchor texts consist of considerably longer token spans than event mentions annotated in gold-standard corpora. This is due to the fact that hyperlink anchor texts are often phrases or entire sentences, opposed to minimum span annotations (see Figure 6). In order to provide rough insights into the proportion of nominalized vs. verbal event mentions in each corpus, we determine the coarse-grained part-of-speech tag of mention heads for each corpus (Figure 5) using CoreNLP. The majority of ECB+ consists of verbal mentions whereas FCC-T and GVC mostly contain nominalized mentions and a certain amount of adjectival mentions. HyperCoref subcorpora again exhibit properties similar to gold-standard corpora. The most frequent WordNet synsets of all event mentions in each gold-standard corpus and of all hyperlink anchor texts in HyperCoref subcorpora are reported in Table 6.

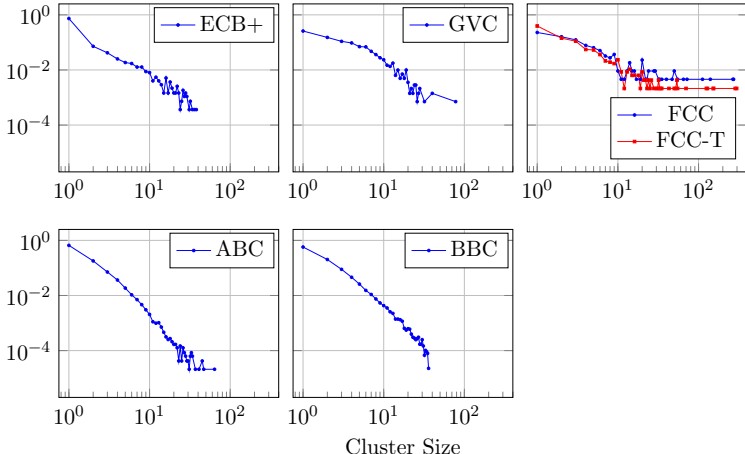

Figure 3: Distribution of cluster sizes by corpus.

| Corpus | Most frequent WordNet synsets |
|---|---|
| ECB+ [Cybulska and Vossen, 2014b] | state.v.01, kill.v.01, die.v.01, earthquake.n.01, announce.v.01, collar.v.01, murder.n.01, death.n.01, shooting.n.01, put.v.01, fire.n.01, hit.v.01, charge.v.01, injure.v.01, make.v.01, take.v.01, attack.n.01, find.v.01, check.v.01, report.v.01 |
| FCC-T [Bugert et al., 2020, 2021] | cup.n.01, tournament.n.01, concluding.s.01, euro.n.01, semi-final, win.v.01, game.n.01, quarter-final, beat.v.01, phase.n.01, match.n.01, victory.n.01, against, defeat.n.01, lose.v.01, round.n.01, sixteen.n.01, semifinal.n.01, championship.n.01, play.v.01 |
| GVC [Vossen et al., 2018] | shooting.n.01, shoot.v.01, kill.v.01, die.v.01, death.n.01, dead.a.01, wound.n.01, incident.n.01, gunfire.n.01, information_technology.n.01, murder.n.01, injure.v.01, open_fire.v.01, gunshot, fatally.r.01, injury.n.01, accident.n.01, fatal.a.01, travel.v.01, homicide.n.01 |

| Corpus | Most frequent WordNet synsets |
|---|---|
| HYPERCOREF (ours) | |
| abcnews.go.com | state.v.01, announce.v.01, kill.v.01, make.v.01, report.v.01, die.v.01, find.v.01, name.v.01, shooting.n.01, travel.v.01, take.v.01, argument.n.03, death.n.01, have.v.01, interview.n.01, attack.n.01, plan.n.01, collar.v.01, put.v.01, probe.n.01 |
| apnews.com | market.n.01, state.v.01, loss.n.01, win.v.01, victory.n.01, announce.v.01, lose.v.01, plead.v.01, die.v.01, have.v.01, report.v.01, beat.v.01, charge.v.01, make.v.01, take.v.01, travel.v.01, find.v.01, sign.v.01, lawsuit.n.01, probe.n.01 |
| avclub.com | episode.n.01, show.n.01, movie.n.01, report.v.01, make.v.01, week.n.01, announce.v.01, state.v.01, be.v.01, have.v.01, travel.v.01, get.v.01, write.v.01, interview.n.01, play.v.01, series.n.01, take.v.01, one.n.01, season.n.01, dawdler.n.01 |
| bbc.com | win.v.01, defeat.n.01, beat.v.01, join.v.01, lose.v.01, victory.n.01, draw.n.01, sign.v.01, take.v.01, leave.v.01, state.v.01, travel.v.01, loss.n.01, pull.v.01, make.v.01, sack.v.01, have.v.01, appoint.v.01, name.v.01, score.v.01 |
| bleacherreport.com | write.v.01, state.v.01, make.v.01, have.v.01, draft.n.01, ranking.n.01, miller.n.01, player.n.01, travel.v.01, week.n.01, be.v.01, take.v.01, injury.n.01, predict.v.01, win.v.01, game.n.01, team.n.01, one.n.01, get.v.01, prospect.n.01 |
| bostonglobe.com | state.v.01, announce.v.01, report.v.01, travel.v.01, make.v.01, name.v.01, put.v.01, take.v.01, plan.n.01, remark.n.01, kill.v.01, let_go_of.v.01, pass.v.01, ask.v.01, raise.v.01, have.v.01, look.v.02, file.v.01, report.n.01, propose.v.01 |
| businessinsider.com | state.v.01, make.v.01, have.v.01, report.v.01, announce.v.01, be.v.01, travel.v.01, get.v.01, take.v.01, name.v.01, get_down.v.07, put.v.01, write.v.01, $, buy.v.01, come.v.01, establish.v.01, one.n.01, use.v.01, show.v.01 |
| cnbc.com | state.v.01, report.v.01, announce.v.01, make.v.01, have.v.01, be.v.01, fall.v.01, take.v.01, travel.v.01, get.v.01, put.v.01, rise.v.01, net_income.n.01, expect.v.01, get_down.v.07, buy.v.01, raise.v.01, $, hit.v.01, cut.v.01 |
| cnn.com | state.v.01, kill.v.01, announce.v.01, make.v.01, sound.n.01, take.v.01, travel.v.01, have.v.01, die.v.01, attack.n.01, name.v.01, be.v.01, win.v.01, find.v.01, wav, write.v.01, report.v.01, get_down.v.07, put.v.01, narrative.n.01 |
| deadspin.com | make.v.01, have.v.01, state.v.01, travel.v.01, get.v.01, be.v.01, take.v.01, name.v.01, know.v.01, win.v.01, game.n.01, write.v.01, one.n.01, come.v.01, player.n.01, open_fire.v.01, report.v.01, try.v.01, lose.v.01, put.v.01 |
| denverpost.com | state.v.01, have.v.01, make.v.01, announce.v.01, travel.v.01, get_down.v.07, plan.n.01, take.v.01, find.v.01, get.v.01, die.v.01, name.v.01, put.v.01, win.v.01, be.v.01, kill.v.01, collar.v.01, report.v.01, shoot.v.01, file.v.01 |
| espn.com | state.v.01, win.v.01, loss.n.01, week.n.01, make.v.01, have.v.01, sign.v.01, deal.n.01, announce.v.01, victory.n.01, report.v.01, take.v.01, ranking.n.01, travel.v.01, put.v.01, get.v.01, write.v.01, trade.v.01, player.n.01, agree.v.01 |
| formula1.com | take.v.01, make.v.01, state.v.01, Prix, clang.n.01, win.v.01, crash.v.01, have.v.01, put.v.01, punishment.n.01, announce.v.01, race.n.01, complete.v.01, get_down.v.07, spin.v.01, lose.v.01, lap.n.01, come.v.01, get.v.01, qualify.v.01 |
| foxnews.com | state.v.01, report.v.01, announce.v.01, make.v.01, have.v.01, take.v.01, win.v.01, name.v.01, travel.v.01, trump.n.01, poll.n.01, put.v.01, get.v.01, write.v.01, victory.n.01, be.v.01, let_go_of.v.01, interview.n.01, kill.v.01, accuse.v.01 |
| gizmodo.com | make.v.01, state.v.01, have.v.01, get.v.01, be.v.01, announce.v.01, travel.v.01, telephone.n.01, app, use.v.01, report.v.01, come.v.01, one.n.01, take.v.01, year.n.01, see.v.01, establish.v.01, get_down.v.07, know.v.01, look.v.01 |
| huffpost.com | state.v.01, make.v.01, announce.v.01, have.v.01, write.v.01, name.v.01, report.v.01, travel.v.01, take.v.01, be.v.01, come.v.01, put.v.01, die.v.01, post.n.01, interview.n.01, get.v.01, let_go_of.v.01, find.v.01, use.v.01, keep.v.01 |
| independent.co.uk | state.v.01, name.v.01, announce.v.01, kill.v.01, make.v.01, attack.n.01, have.v.01, take.v.01, accuse.v.01, uncover.v.01, victory.n.01, die.v.01, win.v.01, find.v.01, trump.n.01, travel.v.01, claim.v.01, put.v.01, be.v.01, shoot.v.01 |
| kotaku.com | game.n.01, make.v.01, state.v.01, announce.v.01, get.v.01, have.v.01, be.v.01, travel.v.01, report.v.01, one.n.01, come.v.01, dawdler.n.01, play.v.01, look.v.01, write.v.01, take.v.01, see.v.01, show.v.01, uncover.v.01, manner.n.01 |
| lawandcrime.com | state.v.01, report.v.01, make.v.01, write.v.01, file.v.01, note.v.01, have.v.01, accuse.v.01, law.n.01, travel.v.01, argue.v.01, claim.v.01, lawsuit.n.01, collar.v.01, name.v.01, announce.v.01, action.v.01, be.v.01, case.n.01, charge.v.01 |
| marketwatch.com | state.v.01, fall.v.01, rise.v.01, report.v.01, %, see.v.01, make.v.01, stock.n.01, have.v.01, net_income.n.01, be.v.01, market.n.01, close.v.01, get.v.01, announce.v.01, travel.v.01, take.v.01, drop.v.01, cut.v.01, show.v.01 |
| metro.us | state.v.01, get.v.01, make.v.01, be.v.01, have.v.01, man.n.01, find.v.01, take.v.01, travel.v.01, announce.v.01, collar.v.01, day.n.01, die.v.01, season.n.01, name.v.01, shoot.v.01, report.v.01, shooting.n.01, put.v.01, let_go_of.v.01 |
| mirror.co.uk | state.v.01, win.v.01, have.v.01, make.v.01, uncover.v.01, travel.v.01, put.v.01, be.v.01, write.v.01, take.v.01, defeat.n.01, leave.v.01, lose.v.01, claim.v.01, find.v.01, desire.v.01, show.v.01, beat.v.01, man.n.01, die.v.01 |
| nbc.com | state.v.01, announce.v.01, kill.v.01, make.v.01, report.v.01, write.v.01, have.v.01, find.v.01, die.v.01, travel.v.01, take.v.01, name.v.01, poll.n.01, be.v.01, ET, attack.n.01, get_down.v.07, put.v.01, collar.v.01, charge.v.01 |
| newrepublic.com | write.v.01, state.v.01, argue.v.01, note.v.01, make.v.01, Vinik, Leber, have.v.01, indicate.v.02, report.v.01, explain.v.01, argument.n.03, propose.v.01, be.v.01, plan.n.01, travel.v.01, take.v.01, Beutler, get.v.01, cohn.n.01 |
| newsweek.com | state.v.01, report.v.01, name.v.01, announce.v.01, kill.v.01, make.v.01, have.v.01, Newsweek, attack.n.01, take.v.01, travel.v.01, accuse.v.01, claim.v.01, knock.v.06, trump.n.01, find.v.01, interview.n.01, propose.v.01, let_go_of.v.01, die.v.01 |
| politico.com | state.v.01, report.v.01, announce.v.01, besides.r.02, Obama, make.v.01, address.n.03, name.v.01, poll.n.01, argument.n.03, plan.n.01, bill.n.01, take.v.01, travel.v.01, $, democrat.n.01, raise.v.01, interview.n.01, politician.n.02, put.v.01 |
| rollingstone.com | state.v.01, album.n.01, tour.n.01, rock.n.01, interview.n.01, report.v.01, die.v.01, video.n.01, announce.v.01, perform.v.01, let_go_of.v.01, song.n.01, make.v.01, show.n.01, talk.v.02, write.v.01, death.n.01, cancel.v.01, concert.n.01, performance.n.01 |
| seattletimes.com | state.v.01, announce.v.01, plan.n.01, travel.v.01, make.v.01, report.v.01, find.v.01, get_down.v.07, approve.v.01, close.v.01, pass.v.01, shoot.v.01, name.v.01, put.v.01, year.n.01, have.v.01, file.v.01, take.v.01, die.v.01, propose.v.01 |

| Corpus | Most frequent WordNet synsets |
|---|---|
| technologyreview.com | make.v.01, be.v.01, use.v.01, state.v.01, technology.n.01, have.v.01, get.v.01, take.v.01, announce.v.01, report.v.01, show.v.01, travel.v.01, develop.v.01, write.v.01, system.n.01, construct.v.01, establish.v.01, work.v.01, one.n.01, test.v.01 |
| theepochtimes.com | state.v.01, announce.v.01, report.v.01, collar.v.01, times.n.01, make.v.01, find.v.01, name.v.01, travel.v.01, let_go_of.v.01, write.v.01, take.v.01, interview.n.01, have.v.01, confirm.v.01, kill.v.01, charge.v.01, pass.v.01, case.n.01, sign.v.01 |
| theguardian.com | state.v.01, win.v.01, make.v.01, announce.v.01, take.v.01, have.v.01, defeat.n.01, travel.v.01, name.v.01, lose.v.01, kill.v.01, victory.n.01, put.v.01, be.v.01, attack.n.01, uncover.v.01, warn.v.01, report.v.01, accuse.v.01, die.v.01 |
| thehill.com | state.v.01, announce.v.01, name.v.01, report.v.01, write.v.01, let_go_of.v.01, tweet.v.01, poll.n.01, defend.v.01, travel.v.01, back.v.01, take.v.01, pass.v.01, report.n.01, put.v.01, make.v.01, ask.v.01, push.v.01, knock.v.06, show.v.01 |
| theverge.com | announce.v.01, make.v.01, establish.v.01, state.v.01, app, have.v.01, travel.v.01, get.v.01, get_down.v.07, plan.n.01, take.v.01, let_go_of.v.01, use.v.01, come.v.01, put.v.01, service.n.01, report.v.01, uncover.v.01, version.n.01, tablet.n.01 |
| theweek.com | state.v.01, announce.v.01, name.v.01, make.v.01, report.v.01, have.v.01, travel.v.01, claim.v.01, tweet.v.01, take.v.01, propose.v.01, put.v.01, let_go_of.v.01, get_down.v.07, get.v.01, try.v.01, keep.v.01, ask.v.01, look.v.02 |
| usatoday.com | state.v.01, have.v.01, make.v.01, name.v.01, travel.v.01, announce.v.01, be.v.01, die.v.01, take.v.01, get.v.01, kill.v.01, find.v.01, trump.n.01, put.v.01, keep.v.01, confront.v.02, show.v.01, get_down.v.07, shooting.n.01, let_go_of.v.01 |
| vanityfair.com | state.v.01, report.v.01, make.v.01, announce.v.01, hive.n.01, travel.v.01, name.v.01, write.v.01, propose.v.01, have.v.01, reportedly.r.01, take.v.01, claim.v.01, put.v.01, note.v.01, come.v.01, be.v.01, try.v.01, interview.n.01, week.n.01 |
| vox.com | send.v.01, state.v.01, make.v.01, be.v.01, have.v.01, write.v.01, explain.v.01, travel.v.01, get.v.01, day.n.01, name.v.01, report.v.01, thing.n.01, plan.n.01, announce.v.01, put.v.01, one.n.01, use.v.01, take.v.01, episode.n.01 |
| washingtonpost.com | state.v.01, report.v.01, write.v.01, make.v.01, have.v.01, be.v.01, win.v.01, announce.v.01, travel.v.01, take.v.01, name.v.01, put.v.01, get.v.01, loss.n.01, get_down.v.07, show.v.01, note.v.01, find.v.01, kill.v.01, keep.v.01 |
| wired.com | make.v.01, state.v.01, get.v.01, use.v.01, have.v.01, travel.v.01, take.v.01, announce.v.01, be.v.01, system.n.01, get_down.v.07, establish.v.01, construct.v.01, put.v.01, year.n.01, plan.n.01, work.v.01, name.v.01, write.v.01, keep.v.01 |
| wsj.com | state.v.01, article.n.01, report.v.01, fall.v.01, rise.v.01, agree.v.01, plan.v.01, make.v.01, take.v.01, put.v.01, raise.v.01, get_down.v.07, column.n.01, have.v.01, travel.v.01, announce.v.01, buy.v.01, name.v.01, cut.v.01, expect.v.01 |

Table 6: The 20 most frequent WordNet synsets of event mentions per corpus, ordered from most to least frequent. Synsets are obtained by (1) dependency parsing sentences, (2) reducing mention spans to the phrase head of their span, (3) looking up the most frequent WordNet synset of each span (using the lemma as a fallback if no synset is available). Synsets appearing in the top 20 of at least 75% of all corpora are marked in gray.

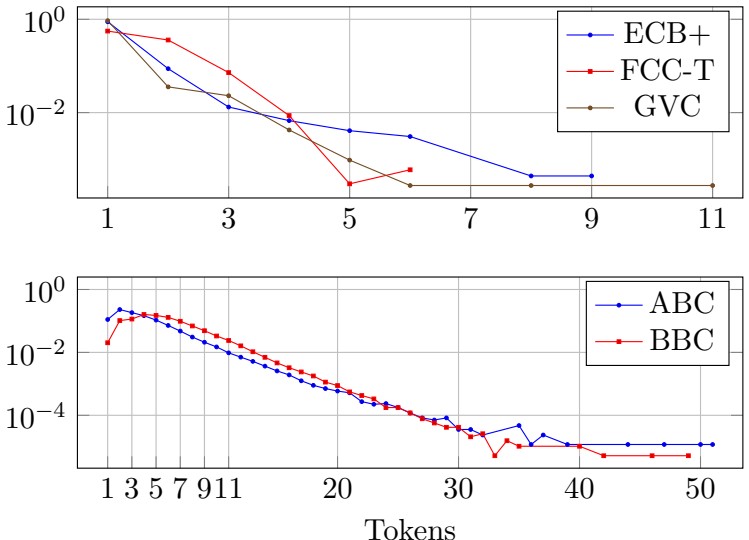

Figure 4: Distribution of mention span lengths by corpus.

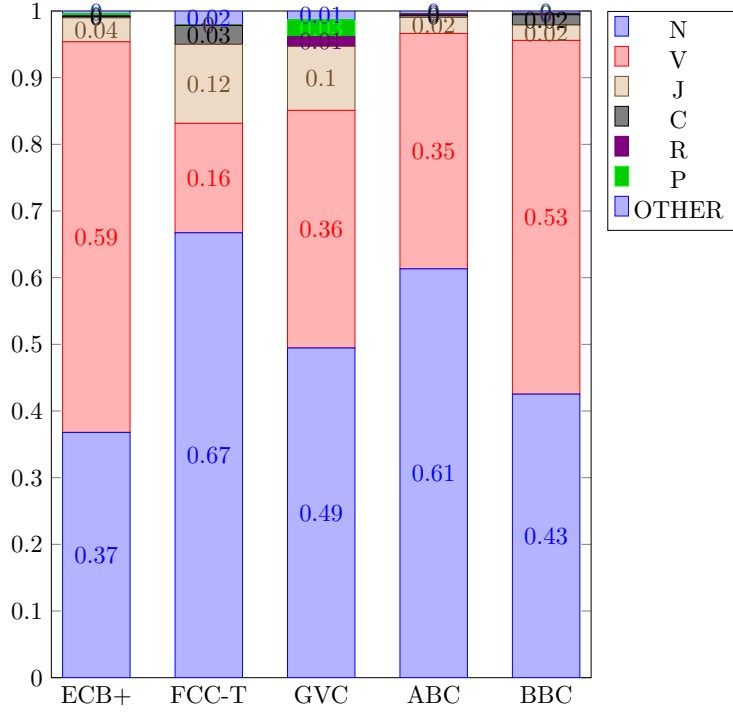

Figure 5: Distribution of coarse-grained part-of-speech tags for mention heads.

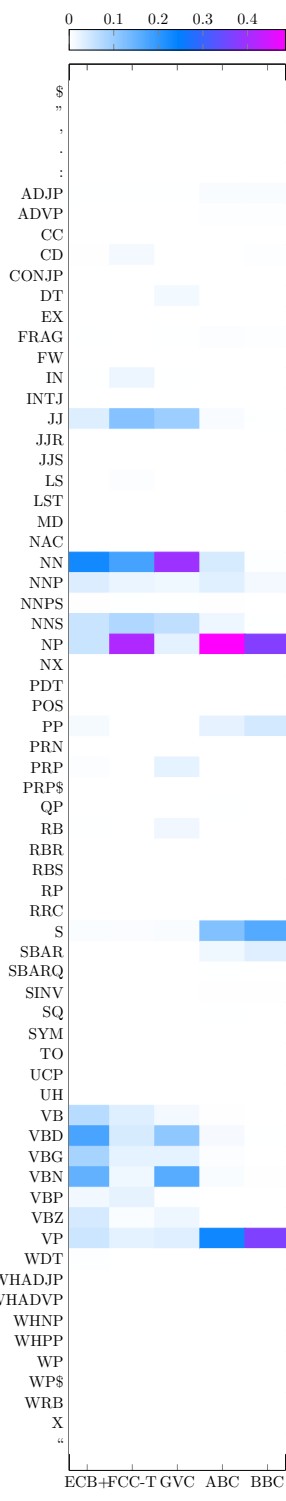

Figure 6: Relative frequency of constituency phrase types of mentions.

| Corpus | Documents | | Mentions | | Events | |
|---|---|---|---|---|---|---|
| | Amount | Publ. date known | Amount[1] | Unique verbs | Clusters[1] | Singletons |
| ● ECB+ [Cybulska and Vossen, 2014b] | 1.0 k | 53.2 %[2] | 4.8 k | 573 | 0.7 k | 2.0 k |
| ● FCC-T [Bugert et al., 2020, 2021] | 0.5 k | 100.0 % | 3.2 k | 91 | 0.3 k | 0.2 k |
| ● GVC [Vossen et al., 2018] | 0.5 k | 100.0 % | 6.9 k | 71 | 1.0 k | 0.4 k |
| ● WEC [Eirew et al., 2021] | 37.1 k | n/a[3] | 42.8 k | —[4] | 6.7 k | 0.9 k |
| ● HyperCoref (ours) | 2046.0 k | 76.7 % | 2743.7 k | 4071 | 810.6 k | 1564.4 k |
| businessinsider.com | 221.1 k | 89.6 % | 337.6 k | 2901 | 95.3 k | 124.8 k |
| huffpost.com | 168.0 k | 100.0 % | 208.1 k | 2866 | 62.4 k | 121.2 k |
| mirror.co.uk | 165.2 k | 100.0 % | 213.4 k | 2570 | 52.6 k | 104.7 k |
| theguardian.com | 165.1 k | 99.9 % | 303.9 k | 2916 | 82.7 k | 112.2 k |
| washingtonpost.com | 142.2 k | 99.4 % | 240.8 k | 2839 | 70.5 k | 108.8 k |
| cnbc.com | 132.0 k | 93.0 % | 154.2 k | 2223 | 48.5 k | 83.3 k |
| marketwatch.com | 104.3 k | 83.3 % | 202.3 k | 2265 | 56.8 k | 82.2 k |
| bbc.com | 98.2 k | 98.4 % | 142.1 k | 1520 | 37.6 k | 50.7 k |
| foxnews.com | 65.6 k | 53.0 % | 46.4 k | 1933 | 16.1 k | 73.0 k |
| gizmodo.com | 65.6 k | 0.1 % | 85.4 k | 2186 | 24.8 k | 40.0 k |
| nbc.com | 58.6 k | 30.9 % | 77.1 k | 2139 | 23.3 k | 46.0 k |
| bleacherreport.com | 51.1 k | 0.2 % | 21.6 k | 1522 | 8.2 k | 46.3 k |
| kotaku.com | 50.5 k | 0.1 % | 52.5 k | 1763 | 17.6 k | 33.0 k |
| usatoday.com | 47.4 k | 99.8 % | 84.8 k | 2193 | 25.4 k | 53.8 k |
| cnn.com | 42.8 k | 73.2 % | 24.4 k | 1874 | 9.9 k | 60.5 k |
| newsweek.com | 41.1 k | 100.0 % | 56.0 k | 1864 | 17.9 k | 28.1 k |
| abcnews.go.com | 38.4 k | 0.0 % | 83.4 k | 1807 | 46.2 k | 31.4 k |
| rollingstone.com | 35.0 k | 100.0 % | 41.6 k | 1397 | 14.2 k | 27.9 k |
| denverpost.com | 34.2 k | 98.4 % | 46.2 k | 1888 | 14.5 k | 33.9 k |
| thehill.com | 32.6 k | 99.6 % | 40.1 k | 1410 | 12.6 k | 23.5 k |
| apnews.com | 29.3 k | 92.5 % | 25.8 k | 1626 | 9.0 k | 30.8 k |
| deadspin.com | 28.9 k | 0.1 % | 32.1 k | 1880 | 10.8 k | 20.1 k |
| avclub.com | 26.6 k | 19.2 % | 22.0 k | 1335 | 8.7 k | 32.0 k |
| theverge.com | 26.1 k | 99.0 % | 39.3 k | 1468 | 11.4 k | 14.5 k |
| theepochtimes.com | 22.8 k | 99.3 % | 20.5 k | 1310 | 7.0 k | 18.7 k |
| independent.co.uk | 21.5 k | 99.2 % | 18.0 k | 1405 | 6.8 k | 20.8 k |
| wsj.com | 20.5 k | 0.0 % | 19.3 k | 1801 | 7.6 k | 46.5 k |
| metro.us | 17.2 k | 62.4 % | 7.8 k | 1186 | 3.2 k | 17.0 k |
| politico.com | 14.4 k | 27.4 % | 15.0 k | 1343 | 5.4 k | 12.3 k |
| seattletimes.com | 14.3 k | 99.9 % | 25.1 k | 1446 | 7.0 k | 10.2 k |
| vanityfair.com | 10.6 k | 0.5 % | 16.0 k | 1412 | 4.8 k | 5.7 k |
| wired.com | 9.4 k | 0.0 % | 25.3 k | 1496 | 6.3 k | 6.9 k |
| espn.com | 9.2 k | 0.2 % | 10.4 k | 1114 | 3.4 k | 10.7 k |
| theweek.com | 7.3 k | 92.3 % | 7.0 k | 1040 | 2.6 k | 6.2 k |
| vox.com | 7.0 k | 26.0 % | 3.5 k | 816 | 1.4 k | 10.3 k |
| technologyreview.com | 6.7 k | 90.6 % | 8.1 k | 866 | 2.6 k | 5.3 k |
| lawandcrime.com | 6.2 k | 0.0 % | 7.2 k | 785 | 2.4 k | 3.3 k |
| newrepublic.com | 4.3 k | 100.0 % | 3.1 k | 731 | 1.2 k | 3.5 k |
| formula1.com | 3.4 k | 0.1 % | 5.6 k | 641 | 1.7 k | 2.5 k |
| bostonglobe.com | 1.4 k | 99.9 % | 0.8 k | 533 | 0.3 k | 2.1 k |

[1] Excluding singletons.
[2] According to re-annotations of Bugert et al. [2021].
[3] Encyclopedic articles do not have a publication date by design.
[4] Did not determine.

Table 4: Size comparison between CDCR corpora with gold annotations and the proposed HyperCoref corpus with silver annotations.

## A.2 Qualitative Analysis Details

Table 7 reports our classification of 300 HyperCoref mentions as to whether their hyperlink anchor text or surrounding sentence refers to event-like content. Invalid event references account for cases where a hyperlink qualifies as an event mention but the referenced article covers a different event, and for hyperlinks which do refer to the correct event but are not related to their surrounding document.

Given a hyperlink anchor text, we analyze where mentions of action, participants, time and location are located in the surrounding document (see Table 10). Actions are predominantly found inside the anchor text and the major participants mostly in the surrounding sentence. Times and locations of events need to be determined from the document context more frequently.

As demonstrated in Table 9, the vast majority of the analyzed event mentions in HyperCoref references past events.

We examine the relation between the event(s) mentioned by the sentence surrounding a hyperlink and the main event the article referenced by this hyperlink is reporting about. Table 8 shows that the majority of links refer to the main event, while a smaller proportion references a subevent of the article's main event. Several such examples are A1, B1, B2 in Table 11. Note that B1 contains mentions of the main event (the interview) and a sub-event (Biden saying he takes responsibility), with the hyperlink being placed on the sub-event mention. We also observed the opposite case in which the event mentioned in the sentence containing a hyperlink encompasses the event in the referenced article. An example is A2 in Table 11. Here, a hyperlink refers to a mass shooting, with the referenced article reporting about a recent aspect of this crime (the indictment of the offender). Such cases tend to happen when writers merely provide context on a topic, rather than citing a specific incident.

Significant real-world events need little information to be recognizable – two prime examples are the 2001 terrorist attacks on New York's World Trade Center which are oftentimes referenced only by their date ("9/11") or the US Independence Day which is celebrated on the 4th of July every year and therefore uniquely recognizable by its date. We observed similar cases in HyperCoref during a pre-study: In Table 11, example F1 refers to a motor race which took place in 2019 in Canada involving driver Sebastian Vettel, yet there is no explicit lexical trigger for the event action since readers will infer the action from the document context. Instead, "Canada", being the location of the event, takes over the role of the lexical trigger. Event mentions of this kind have so far only been considered for historically significant events (such as 9/11 or World War II) [Cybulska and Vossen, 2014a], hence it would be vital to retain these in HyperCoref. A different case is example F2: in this sentence, "in Canada" refers to the country of a race event, but was chosen as the anchor text for reasons of emphasis. Yet another special case is shown in example B3 which cites a football trainer's previous employment period. Leading a team until a certain date implicitly references a contract expiry event, which here is marked with a hyperlink to an article discussing said event.

Identifying such edge cases in order to correct anchor texts which are misplaced (from a linguistic point of view) as in example F2, while keeping cases like examples F1 and B3 unchanged may improve the quality of the data considerably. However, we expect reliable identification of these cases to be very difficult. Given a risk of introducing biases in the

data through machine learning based filtering techniques, as well as the computational cost of applying such a solution to large volumes of data, we decided against filtering techniques going beyond the rule-based approaches described in Section 3.1. The detection and correct resolution of the previously mentioned action-less event mentions in particular may however pose an interesting topic for future research.

| Anchor text and surrounding sentence contain ... | ABC | BBC |
|---|---|---|
| in-context event reference | 74 | 136 |
| invalid event reference | 28 | 11 |
| no event reference | 47 | 4 |

Table 7: Manual classification of 300 mentions taken from HyperCoref.

| Relation | ABC | BBC |
|---|---|---|
| main event | 58 | 132 |
| sub-event | 18 | 18 |
| super-event | 6 | 7 |

Table 8: Relation between events mentioned inside or surrounding hyperlink anchor texts and the main event of the referenced article. Multiple relations possible. Based on all "in-context event reference" mentions from Table 7.

| Relation | ABC | BBC |
|---|---|---|
| past event | 94.6 % | 91.9 % |
| future/irrealis event | 5.4 % | 8.1 % |

Table 9: Percentage of event mentions referencing past vs. future events. Based on all "in-context event reference" mentions from Table 7.

| Position | A | P | T | L |
|---|---|---|---|---|
| inside anchor text | 65.7 % | 31.0 % | 20.0 % | 9.0 % |
| same sentence | 32.4 % | 61.0 % | 26.2 % | 8.1 % |
| previous/next sentence | 1.0 % | 3.8 % | 3.8 % | 3.3 % |
| elsewhere/nowhere | 1.0 % | 4.3 % | 50.0 % | 79.5 % |

Table 10: Location of event components (**A**ction, **P**articipants, **T**ime, **L**ocation) surrounding anchor texts. Based on all "in-context event reference" mentions from Table 7.

| Referenced article | | Article excerpts with hyperlink reference to the article shown left |
|---|---|---|
| Russia's Maria Sharapova recovered from a nervous start to beat Romanian fourth seed Simona Halep and win her first Madrid Open title. | **A1** | Sharapova remains an obstacle she has yet to overcome, however, with the Russian having won all three of their previous matches. Halep draws hope from the fact that she has improved each time, pushing the former world number one to three sets in Madrid recently. |
| In their first interview since announcing his candidacy, former Vice President Joe Biden and Dr. Jill Biden sat down with ABC's "Good Morning America" co-anchor Robin Roberts and addressed issues from Biden's past that have drawn criticism. | **B1** | In an interview with ABC's "Good Morning America" co-anchor Robin Roberts Monday, Biden said he takes responsibility for Hill's treatment in 1991 when she testified before the Senate Judiciary committee during Supreme Court Justice Clarence Thomas's confirmation hearing. |
| | **B2** | Biden also tried to position himself as the antithesis of President Donald Trump. |
| Patrick Crusius, the alleged gunman in the El Paso shooting, has been indicted for capital murder by a grand jury in Texas. | **A2** | Sen. Ted Cruz, R-Texas, said there have been "too damn many" mass shootings in Texas, but claimed that gun control proposals from Democrats would not have stopped the recent mass shootings in his home state. |
| Lewis Hamilton secured a record-breaking seventh win at the 2019 Canadian Grand Prix, after a penalty for Sebastian Vettel, who finished first on the road, demoted the German to second in the standings. | **F1** | Vettel, on other hand, could not muster a smile. Since Canada, Sebastian seems to have been struggling more and more, and at Silverstone those woes deepened further. |
| n/a | **F2** | A hint of tension between the Force India drivers had been seen at the previous race in Canada, when Perez had refused to let Ocon past to try and attack Ricciardo. |
| The Ghana Football Association (GFA) says it has parted company with national team coach Kwesi Appiah by mutual consent. | **B3** | He replaces Avram Grant who stepped down as coach after the 2017 Africa Cup of Nations. It is a second stint in charge for Appiah, who led the Black Stars from 2012 until 2014. Since leaving the Black Stars following a poor World Cup campaign, he has been coaching Sudanese side Al Khartoum. |

Table 11: Notable example mentions from HyperCoref.

### A.3 Details on the Application of Cattan et al. [2021]

We use a batch size of 128 (Cattan et al. use 32), with a learning rate of `5e-5` (Cattan et al. use `1e-4`). Compared to Cattan et al. where each model was trained for a fixed number of 10 epochs, we train for up to 100 epochs, using early stopping on the development split with a patience of 7 epochs. We optimize the clustering hyperparameter $\tau \in \{0.5, 0.55, 0.6, 0.65, 0.7\}$ on the respective development split for the model which achieved the lowest loss on the development split during training.

Having faced out-of-memory errors when training on topics larger than roughly 50 documents, we used the KaHyPar framework [Schlag, 2020] to partition HYPERCOREF data into pseudo-topics of 50 documents each (where the number of hyperlinks lost in the partitioning process is minimized).

**Coreference Resolution**   The following information only concerns coreference resolution experiments.

There are diminishing returns in generating all possible coreferring mentions pairs at training time, particularly in corpora with large clusters such as FCC-T which lead to the generation of many similar pairs [Bugert et al., 2021]. To address this, we sample at most $6 \cdot \sqrt{n}$ coreferring mention pairs for a given cluster of $n$ mentions. Regarding the population of non-coreferring pairs, Cattan et al. train with up to 20 times as many non-coreferring pairs per topic as there are coreferring pairs.

We reduced this ratio to 15 to speed up the training process. GVC has the unique property of consisting of a single topic and many small clusters, leading to a highly skewed ratio of non-coreferring pairs to coreferring pairs in the training split (GVC: 498:1, ECB+: 20:1, FCC-T: 19:1). After observing strong model bias towards predictions of the non-coreferring class on the GVC development set using a ratio of 15, we reduced the ratio of non-coreferring to coreferring training pairs to 5 for all $S_{sg}$ experiments (partially) trained on GVC.

Analogous to the experiments of Cattan et al., all coreference resolution experiment results are reported using "predicted topics" following Barhom et al. [2019]. This entails preclustering the set of test documents using TF-IDF, generating separate event coreference clusters within each of these document clusters, merging the predictions for each document cluster into a single meta-document, followed by the computation of coreference resolution metrics from this meta-document. For FCC-T and GVC, we use document preclusterings created by Bugert et al. [2021] in the above manner.

We include singletons throughout our experiments. For the lemma and lemma-$\delta$ baselines, we use the implementation from Bugert et al. [2021].

Apart from the differences mentioned above, please note that our in-domain ECB+ results in the $S_{gg}$ scenario are different from the results reported by Cattan et al. [2021] since their model was trained on event and entity annotations. To ensure comparability with FCC-T and GVC (which do not offer entity coreference annotations), we only make use of event annotations for ECB+.

### A.4 Full Experiment Results

We here report the performance of two CDCR baselines, as well as the full set of coreference resolution metrics for the experiments conducted. Furthermore, we tested the use of HyperCoref for event mention detection.

#### A.4.1 Coreference Resolution

We measure coreference resolution performance with the MUC [Vilain et al., 1995], CEAF$_e$ [Luo, 2005], B$^3$ [Bagga and Baldwin, 1998], CoNLL F1 [Pradhan et al., 2012] and LEA [Moosavi and Strube, 2016] metrics. We use the scorer implementation from `https://github.com/ns-moosavi/coval`.

**Baselines**   We report two common CDCR baselines. The `lemma` baseline clusters all event mentions with the same head lemma together. `lemma-`$\delta$ is a trainable variant of `lemma` which restricts merging to document pairs which exceed a TF-IDF cosine similarity of $\delta$ [Upadhyay et al., 2016]. We train $\delta$ on the gold development split of each respective corpus.

**Results**   Tables 12 to 16 report the full P/R/F1 scores of the coreference resolution experiments reported in Section 4 for each of these metrics respectively.

With respect to baselines, the HyperCoref models outperform the baselines in both the $S_{ss}$ and $S_{sg}$ scenarios. The lemma-$\delta$ baseline adapts to the distribution of lexically similar mentions in clusters of documents with similar content. This is a highly corpus-dependent property, explaining the baseline's strong in-domain performance (which surpasses the Cattan et al. [2021] system in the $S_{sg}$ scenario in two occasions) while it leads to significantly worse performance on unseen test corpora.

Table 18 reports the best training epochs, clustering thresholds $\tau$ and development set LEA F1 scores of each model and independent trial.

#### A.4.2 Event Mention Detection

We additionally investigate the usefulness of HyperCoref for event mention detection. Of the three CDCR corpora studied, ECB+ is the only corpus allowing general evaluation of event mention detection [Bugert et al., 2021]: in ECB+, mentions were annotated exhaustively per sentence, opposed to FCC-T and GVC where only mentions of specific event types were annotated. The silver event mentions found in HyperCoref are similarly incomplete, as it is unlikely that each event in a given sentence is marked with a separate hyperlink. We therefore expect low recall for models trained on HyperCoref when applied on ECB+.

To measure event detection performance, we train the full CDCR system of Cattan et al. [2021] (training mode "e2e") for the three previously mentioned data scarcity scenarios and test these models on ECB+. Included in the comparison is Reimers [2018] who trained a dedicated event mention detection system using a BiLSTM-CRF architecture [Huang et al., 2015].

The results are shown in Table 17. All models incorporating HyperCoref data exhibit lower precision and much lower recall than the $S_{gg}$ model trained entirely on gold data, partially confirming our expectations. At the same time, Reimers [2018] significantly outperforms Cattan et al. [2021], which most likely stems from its CRF component which

| Scen. | System | Train | Dev | ECB+ | | | FCC-T | | | GVC | | | H. Mean |
| --- | --- | --- | --- | --- | --- | --- | --- | --- | --- | --- | --- | --- | --- |
| | | | | P | R | F1 | P | R | F1 | P | R | F1 | F1 |
| $S_{gg}$ | Cattan et al. | ECB+ | ECB+ | 85.2 ± 1.3 | 75.0 ± 4.4 | 79.7 ± 1.9 | 68.8 ± 0.4 | 70.9 ± 1.8 | 69.9 ± 0.8 | 78.9 ± 0.8 | 75.9 ± 4.4 | 77.3 ± 2.1 | 75.40 |
| | | with ABC | ECB+ | 85.4 ± 1.1 | 66.3 ± 6.9 | 74.5 ± 4.2 | 69.2 ± 2.5 | 59.7 ± 4.7 | 64.0 ± 1.9 | 79.4 ± 1.5 | 70.4 ± 5.8 | 74.6 ± 2.8 | 70.67 |
| | | with BBC | ECB+ | 86.3 ± 0.8 | 66.3 ± 0.7 | 75.0 ± 0.7 | 72.7 ± 2.5 | 61.3 ± 2.0 | 66.5 ± 1.0 | 78.1 ± 0.4 | 75.0 ± 1.2 | 76.5 ± 0.6 | 72.39 |
| | | FCC-T | FCC-T | 66.4 ± 8.4 | 65.4 ± 16.5 | 64.3 ± 4.7 | 84.9 ± 2.8 | 44.1 ± 1.4 | 58.0 ± 1.8 | 80.1 ± 1.4 | 48.7 ± 10.0 | 60.1 ± 7.1 | 60.69 |
| | | with ABC | FCC-T | 65.2 ± 3.6 | 63.9 ± 1.8 | 64.5 ± 2.3 | 85.7 ± 0.5 | 48.1 ± 7.2 | 61.4 ± 5.7 | 79.3 ± 1.0 | 30.7 ± 3.5 | 44.2 ± 3.6 | 55.13 |
| | | with BBC | FCC-T | 66.6 ± 3.1 | 33.3 ± 18.9 | 42.5 ± 18.1 | 62.9 ± 5.4 | 35.8 ± 18.1 | 44.4 ± 16.3 | 76.4 ± 6.4 | 8.1 ± 1.2 | 14.6 ± 2.1 | 26.19 |
| | | GVC | GVC | 66.9 ± 4.6 | 75.8 ± 4.1 | 70.9 ± 0.9 | 68.2 ± 0.2 | 70.9 ± 2.9 | 69.5 ± 1.5 | 84.5 ± 0.7 | 41.2 ± 5.4 | 55.2 ± 4.8 | 64.37 |
| | | with ABC | GVC | 63.7 ± 2.1 | 83.2 ± 0.9 | 72.1 ± 1.0 | 67.8 ± 0.1 | 72.3 ± 1.1 | 70.0 ± 0.5 | 82.6 ± 1.0 | 50.5 ± 1.6 | 62.7 ± 1.5 | 68.02 |
| | | with BBC | GVC | 67.9 ± 5.9 | 81.4 ± 8.0 | 73.6 ± 0.3 | 68.2 ± 0.4 | 68.4 ± 7.3 | 68.2 ± 3.9 | 83.3 ± 1.5 | 42.7 ± 7.9 | 56.2 ± 7.2 | 65.16 |
| $S_{sg}$ | Cattan et al. | ABC | ECB+ | 76.2 ± 2.0 | 80.7 ± 3.7 | 78.3 ± 0.8 | 67.5 ± 0.8 | 71.2 ± 1.3 | 69.3 ± 0.2 | 79.7 ± 0.8 | 66.6 ± 7.4 | 72.4 ± 4.0 | 73.15 |
| | | | FCC-T | 84.5 ± 2.0 | 65.6 ± 8.0 | 73.6 ± 4.5 | 67.9 ± 0.3 | 65.8 ± 1.4 | 66.8 ± 0.8 | 79.6 ± 0.8 | 38.7 ± 15.3 | 51.0 ± 14.2 | 62.29 |
| | | | GVC | 84.8 ± 1.0 | 66.2 ± 3.3 | 74.3 ± 1.8 | 68.2 ± 0.9 | 64.5 ± 3.9 | 66.2 ± 1.8 | 79.1 ± 0.4 | 39.2 ± 0.9 | 52.4 ± 0.8 | 62.96 |
| | | BBC | ECB+ | 73.1 ± 2.1 | 82.4 ± 1.1 | 77.4 ± 0.8 | 76.9 ± 0.5 | 64.2 ± 2.2 | 70.0 ± 1.2 | 78.8 ± 0.6 | 69.0 ± 3.2 | 73.6 ± 2.1 | 73.54 |
| | | | FCC-T | 70.5 ± 1.5 | 85.3 ± 2.3 | 77.1 ± 0.6 | 75.4 ± 1.4 | 67.2 ± 2.4 | 71.0 ± 0.9 | 79.6 ± 0.3 | 73.0 ± 2.2 | 76.2 ± 1.3 | 74.67 |
| | | | GVC | 73.5 ± 2.0 | 81.6 ± 1.0 | 77.3 ± 0.8 | 77.3 ± 0.7 | 63.8 ± 1.4 | 69.9 ± 1.1 | 79.5 ± 0.5 | 68.2 ± 1.6 | 73.4 ± 1.1 | 73.41 |
| | lemma-$\delta$ | n/a | ECB+ | 79.3 | 69.3 | 74.0 | 65.2 | 51.2 | 57.4 | 62.5 | 53.6 | 57.7 | 62.16 |
| | | | FCC-T | 69.9 | 69.6 | 69.7 | 66.1 | 58.5 | 62.1 | 52.0 | 57.7 | 54.7 | 61.56 |
| | | | GVC | 83.2 | 63.5 | 72.0 | 62.4 | 34.3 | 44.3 | 75.8 | 50.9 | 60.9 | 56.73 |
| $S_{ss}$ | Cattan et al. | ABC | ABC | 75.2 ± 2.6 | 81.8 ± 4.5 | 78.3 ± 0.8 | 67.6 ± 0.8 | 71.8 ± 1.6 | 69.6 ± 0.5 | 79.3 ± 1.2 | 69.0 ± 7.7 | 73.6 ± 4.2 | 73.66 |
| | | BBC | BBC | 66.1 ± 2.0 | 89.7 ± 1.3 | 76.1 ± 1.0 | 72.0 ± 0.3 | 72.1 ± 0.2 | 72.0 ± 0.2 | 79.0 ± 0.4 | 79.1 ± 1.7 | 79.1 ± 0.6 | 75.62 |
| | lemma | n/a | n/a | 59.7 | 69.7 | 64.3 | 66.2 | 58.8 | 62.3 | 52.1 | 57.9 | 54.8 | 60.18 |

Table 12: MUC scores

| Scen. | System | Train | Dev | ECB+ | | | FCC-T | | | GVC | | | H. Mean |
|---|---|---|---|---|---|---|---|---|---|---|---|---|---|
| | | | | P | R | F1 | P | R | F1 | P | R | F1 | F1 |
| $S_{gg}$ | Cattan et al. | ECB+ | ECB+ | 72.4 ± 3.6 | 82.6 ± 1.8 | 77.1 ± 1.2 | 26.9 ± 2.5 | 22.3 ± 2.3 | 24.3 ± 0.8 | 37.1 ± 2.1 | 42.7 ± 9.8 | 39.4 ± 5.2 | 37.73 |
| | | with ABC | ECB+ | 65.5 ± 4.6 | 82.9 ± 1.3 | 73.1 ± 2.4 | 18.9 ± 1.6 | 31.9 ± 7.0 | 23.4 ± 0.9 | 35.1 ± 3.2 | 49.8 ± 8.0 | 40.7 ± 1.3 | 37.04 |
| | | with BBC | ECB+ | 65.5 ± 0.7 | 83.9 ± 0.7 | 73.5 ± 0.7 | 20.9 ± 1.2 | 38.0 ± 4.2 | 26.9 ± 1.2 | 36.4 ± 0.7 | 42.1 ± 2.2 | 39.0 ± 1.0 | 39.26 |
| | | FCC-T | FCC-T | 56.8 ± 4.1 | 54.4 ± 22.8 | 53.2 ± 11.9 | 19.2 ± 0.1 | 67.9 ± 0.8 | 29.9 ± 0.1 | 24.4 ± 4.3 | 60.4 ± 3.3 | 34.5 ± 3.6 | 36.93 |
| | | with ABC | FCC-T | 55.3 ± 3.4 | 56.6 ± 5.9 | 55.9 ± 4.5 | 20.5 ± 2.9 | 67.2 ± 2.6 | 31.4 ± 3.3 | 16.9 ± 0.8 | 57.5 ± 0.5 | 26.1 ± 0.9 | 34.07 |
| | | with BBC | FCC-T | 58.9 ± 11.4 | 63.6 ± 20.6 | 58.4 ± 5.4 | 19.6 ± 4.2 | 57.6 ± 6.7 | 28.9 ± 3.8 | 27.0 ± 15.3 | 56.1 ± 4.7 | 34.3 ± 12.6 | 37.09 |
| | | GVC | GVC | 64.5 ± 2.1 | 53.6 ± 10.0 | 58.2 ± 5.6 | 21.5 ± 3.0 | 16.8 ± 3.0 | 18.6 ± 2.1 | 20.9 ± 2.7 | 62.6 ± 3.0 | 31.3 ± 3.4 | 29.16 |
| | | with ABC | GVC | 66.6 ± 2.3 | 41.9 ± 5.9 | 51.3 ± 5.2 | 25.6 ± 1.0 | 16.6 ± 1.9 | 20.1 ± 1.5 | 25.9 ± 1.7 | 65.4 ± 3.3 | 37.1 ± 2.2 | 31.19 |
| | | with BBC | GVC | 68.6 ± 1.7 | 50.7 ± 16.2 | 57.3 ± 9.4 | 22.6 ± 4.7 | 20.7 ± 6.6 | 20.6 ± 2.2 | 22.2 ± 3.2 | 64.0 ± 1.9 | 32.9 ± 3.7 | 31.12 |
| $S_{sg}$ | Cattan et al. | ABC | ECB+ | 73.2 ± 2.3 | 67.7 ± 4.6 | 70.3 ± 1.4 | 26.3 ± 0.5 | 18.9 ± 4.5 | 21.8 ± 3.2 | 35.3 ± 5.4 | 56.5 ± 5.9 | 43.0 ± 2.3 | 35.99 |
| | | | FCC-T | 65.0 ± 4.6 | 82.1 ± 2.9 | 72.4 ± 1.9 | 24.5 ± 0.2 | 28.5 ± 2.4 | 26.3 ± 1.1 | 20.8 ± 5.5 | 60.1 ± 0.6 | 30.6 ± 6.0 | 35.50 |
| | | | GVC | 65.3 ± 1.8 | 82.6 ± 1.4 | 72.9 ± 0.7 | 23.5 ± 1.7 | 29.7 ± 6.2 | 25.9 ± 1.7 | 20.1 ± 0.2 | 59.8 ± 0.5 | 30.1 ± 0.2 | 35.07 |
| | | BBC | ECB+ | 73.8 ± 1.2 | 62.3 ± 4.3 | 67.5 ± 2.8 | 24.1 ± 0.5 | 45.1 ± 3.0 | 31.4 ± 0.3 | 35.0 ± 2.1 | 51.9 ± 1.5 | 41.7 ± 1.0 | 42.47 |
| | | | FCC-T | 74.7 ± 2.0 | 55.6 ± 4.7 | 63.6 ± 3.1 | 26.0 ± 1.8 | 40.5 ± 3.8 | 31.5 ± 0.5 | 38.0 ± 2.7 | 50.2 ± 0.1 | 43.2 ± 1.7 | 42.48 |
| | | | GVC | 73.4 ± 0.8 | 63.6 ± 4.3 | 68.1 ± 2.8 | 23.8 ± 0.1 | 45.8 ± 1.7 | 31.3 ± 0.5 | 33.8 ± 1.0 | 52.6 ± 0.6 | 41.1 ± 0.7 | 42.27 |
| | lemma-$\delta$ | n/a | ECB+ | 67.1 | 77.3 | 71.8 | 18.0 | 38.3 | 24.4 | 29.3 | 45.8 | 35.8 | 36.21 |
| | | | FCC-T | 68.7 | 68.9 | 68.8 | 19.3 | 31.0 | 23.8 | 28.3 | 16.3 | 20.7 | 28.61 |
| | | | GVC | 63.3 | 81.4 | 71.2 | 15.8 | 53.5 | 24.4 | 27.4 | 62.8 | 38.1 | 36.91 |
| $S_{ss}$ | Cattan et al. | ABC | ABC | 73.8 ± 2.6 | 65.6 ± 6.1 | 69.3 ± 2.2 | 25.6 ± 1.7 | 17.4 ± 5.2 | 20.5 ± 3.9 | 36.8 ± 5.4 | 54.1 ± 5.9 | 43.3 ± 2.2 | 34.76 |
| | | BBC | BBC | 74.7 ± 1.4 | 42.3 ± 5.7 | 53.9 ± 5.0 | 30.9 ± 1.2 | 30.7 ± 0.8 | 30.8 ± 1.0 | 43.8 ± 2.2 | 43.4 ± 2.7 | 43.5 ± 0.6 | 40.54 |
| | lemma | n/a | n/a | 66.3 | 52.8 | 58.8 | 19.4 | 31.0 | 23.9 | 28.6 | 16.0 | 20.5 | 27.87 |

Table 13: CEAF$_e$ scores

| Scen. | System | Train | Dev | ECB+ | | | FCC-T | | | GVC | | | H. Mean |
|---|---|---|---|---|---|---|---|---|---|---|---|---|---|
| | | | | P | R | F1 | P | R | F1 | P | R | F1 | F1 |
| $S_{gg}$ | Cattan et al. | ECB+ | ECB+ | $86.2 \pm 1.7$ | $75.6 \pm 2.7$ | $80.5 \pm 0.8$ | $45.9 \pm 3.6$ | $60.1 \pm 2.3$ | $52.0 \pm 1.6$ | $54.6 \pm 11.1$ | $62.8 \pm 6.4$ | $57.5 \pm 4.5$ | 61.17 |
| | | with ABC | ECB+ | ECB+ | $88.6 \pm 1.8$ | $69.8 \pm 4.0$ | $78.0 \pm 1.9$ | $58.8 \pm 6.2$ | $44.6 \pm 8.2$ | $50.1 \pm 3.0$ | $63.7 \pm 10.1$ | $57.3 \pm 7.5$ | $59.5 \pm 1.5$ | 60.50 |
| | | with BBC | ECB+ | ECB+ | $89.2 \pm 0.8$ | $68.7 \pm 0.6$ | $77.6 \pm 0.4$ | $61.1 \pm 3.9$ | $47.5 \pm 4.5$ | $53.3 \pm 3.0$ | $53.8 \pm 3.5$ | $61.9 \pm 1.3$ | $57.5 \pm 1.4$ | 61.18 |
| | | FCC-T | FCC-T | $71.2 \pm 20.3$ | $68.8 \pm 8.6$ | $67.9 \pm 7.0$ | $84.1 \pm 3.3$ | $41.4 \pm 8.1$ | $55.2 \pm 8.0$ | $82.1 \pm 8.2$ | $40.1 \pm 6.2$ | $53.4 \pm 3.4$ | 58.17 |
| | | with ABC | FCC-T | FCC-T | $75.1 \pm 4.3$ | $67.1 \pm 1.4$ | $70.8 \pm 1.2$ | $85.6 \pm 4.1$ | $44.1 \pm 7.1$ | $57.9 \pm 5.2$ | $90.1 \pm 1.9$ | $30.4 \pm 2.0$ | $45.4 \pm 2.0$ | 56.16 |
| | | with BBC | FCC-T | FCC-T | $77.3 \pm 18.8$ | $68.1 \pm 14.5$ | $69.9 \pm 2.8$ | $78.0 \pm 10.3$ | $44.4 \pm 6.1$ | $56.0 \pm 2.4$ | $79.8 \pm 18.0$ | $41.6 \pm 18.0$ | $51.2 \pm 11.2$ | 58.03 |
| | | GVC | GVC | $69.5 \pm 8.0$ | $74.2 \pm 3.0$ | $71.5 \pm 2.8$ | $40.2 \pm 5.0$ | $57.0 \pm 9.4$ | $46.7 \pm 4.1$ | $87.8 \pm 2.3$ | $36.4 \pm 2.9$ | $51.4 \pm 2.5$ | 54.69 |
| | | with ABC | GVC | GVC | $62.9 \pm 5.0$ | $79.3 \pm 0.9$ | $70.0 \pm 2.8$ | $40.0 \pm 2.5$ | $61.4 \pm 1.9$ | $48.4 \pm 1.3$ | $85.0 \pm 0.7$ | $40.1 \pm 1.0$ | $54.5 \pm 1.0$ | 56.29 |
| | | with BBC | GVC | GVC | $65.8 \pm 13.5$ | $79.5 \pm 7.0$ | $71.0 \pm 4.7$ | $45.5 \pm 8.5$ | $53.0 \pm 12.2$ | $47.6 \pm 0.6$ | $88.0 \pm 3.2$ | $36.5 \pm 5.2$ | $51.4 \pm 4.7$ | 55.00 |
| $S_{sg}$ | Cattan et al. | ABC | ECB+ | $78.5 \pm 4.1$ | $77.2 \pm 4.4$ | $77.7 \pm 0.3$ | $44.7 \pm 4.8$ | $60.1 \pm 1.5$ | $51.1 \pm 2.6$ | $72.8 \pm 9.3$ | $50.5 \pm 6.7$ | $59.0 \pm 1.3$ | 60.74 |
| | | | FCC-T | $89.4 \pm 3.0$ | $66.9 \pm 3.5$ | $76.4 \pm 1.2$ | $53.6 \pm 1.2$ | $52.1 \pm 4.1$ | $52.7 \pm 1.6$ | $88.2 \pm 5.0$ | $33.9 \pm 6.8$ | $48.5 \pm 6.2$ | 56.94 |
| | | | GVC | $89.6 \pm 1.4$ | $66.7 \pm 1.6$ | $76.4 \pm 0.6$ | $54.4 \pm 5.6$ | $53.0 \pm 4.5$ | $53.4 \pm 0.5$ | $87.9 \pm 0.9$ | $33.6 \pm 0.5$ | $48.6 \pm 0.4$ | 57.26 |
| | | BBC | ECB+ | $73.2 \pm 3.3$ | $78.7 \pm 2.2$ | $75.8 \pm 0.9$ | $64.3 \pm 3.2$ | $50.3 \pm 7.6$ | $56.1 \pm 3.3$ | $66.3 \pm 1.6$ | $54.4 \pm 2.6$ | $59.7 \pm 0.9$ | 62.80 |
| | | | FCC-T | $68.1 \pm 3.3$ | $82.1 \pm 2.5$ | $74.4 \pm 1.3$ | $58.2 \pm 5.0$ | $56.8 \pm 7.8$ | $57.0 \pm 2.9$ | $63.1 \pm 1.4$ | $58.4 \pm 3.2$ | $60.6 \pm 1.0$ | 63.18 |
| | | | GVC | $74.2 \pm 3.0$ | $77.7 \pm 1.4$ | $75.9 \pm 0.9$ | $65.0 \pm 2.0$ | $49.9 \pm 6.9$ | $56.2 \pm 3.4$ | $67.5 \pm 0.5$ | $53.7 \pm 1.5$ | $59.8 \pm 0.8$ | 62.90 |
| | lemma-$\delta$ | n/a | ECB+ | 85.3 | 67.8 | 75.5 | 63.6 | 28.6 | 39.4 | 52.3 | 41.9 | 46.6 | 49.93 |
| | | | FCC-T | 72.2 | 67.9 | 70.0 | 59.3 | 33.2 | 42.5 | 18.7 | 44.1 | 26.3 | 39.56 |
| | | | GVC | 90.5 | 65.5 | 75.9 | 77.1 | 19.1 | 30.6 | 78.8 | 40.5 | 53.5 | 46.48 |
| $S_{ss}$ | Cattan et al. | ABC | ABC | $77.0 \pm 4.7$ | $78.5 \pm 5.0$ | $77.6 \pm 0.2$ | $43.0 \pm 5.4$ | $61.1 \pm 2.1$ | $50.3 \pm 2.9$ | $69.7 \pm 9.0$ | $52.6 \pm 6.8$ | $59.3 \pm 1.5$ | 60.45 |
| | | BBC | BBC | $57.1 \pm 5.2$ | $88.2 \pm 1.6$ | $69.2 \pm 3.5$ | $42.9 \pm 4.5$ | $62.8 \pm 4.0$ | $50.8 \pm 3.7$ | $55.2 \pm 2.7$ | $65.2 \pm 2.0$ | $59.7 \pm 0.8$ | 58.95 |
| | lemma | n/a | n/a | 58.1 | 67.9 | 62.6 | 59.3 | 33.3 | 42.6 | 18.4 | 44.2 | 26.0 | 38.51 |

Table 14: B$^3$ scores

| Scen. | System | Train | Dev | ECB+ | FCC-T | GVC | H. Mean |
|---|---|---|---|---|---|---|---|
| $S_{gg}$ | Cattan et al. | ECB+ | ECB+ | $79.1 \pm 1.3$ | $48.7 \pm 0.4$ | $58.1 \pm 2.8$ | 59.54 |
| | | with ABC | ECB+ | $75.2 \pm 2.9$ | $45.8 \pm 1.4$ | $58.3 \pm 0.9$ | 57.38 |
| | | with BBC | ECB+ | $75.4 \pm 0.6$ | $48.9 \pm 1.7$ | $57.7 \pm 0.7$ | 58.77 |
| | | FCC-T | FCC-T | $61.8 \pm 5.4$ | $47.7 \pm 3.3$ | $49.3 \pm 4.7$ | 52.24 |
| | | with ABC | FCC-T | $63.8 \pm 2.7$ | $50.2 \pm 4.6$ | $38.6 \pm 2.1$ | 48.78 |
| | | with BBC | FCC-T | $63.2 \pm 5.5$ | $48.2 \pm 3.6$ | $46.5 \pm 17.1$ | 51.66 |
| | | GVC | GVC | $66.9 \pm 3.0$ | $45.0 \pm 2.0$ | $46.0 \pm 3.6$ | 50.93 |
| | | with ABC | GVC | $64.5 \pm 3.0$ | $46.2 \pm 0.7$ | $51.4 \pm 1.6$ | 53.00 |
| | | with BBC | GVC | $67.3 \pm 4.6$ | $45.5 \pm 1.4$ | $46.8 \pm 5.2$ | 51.54 |
| $S_{sg}$ | Cattan et al. | ABC | ECB+ | $75.4 \pm 0.3$ | $47.4 \pm 1.9$ | $58.1 \pm 2.5$ | 58.17 |
| | | | FCC-T | $74.2 \pm 2.5$ | $48.6 \pm 0.3$ | $43.4 \pm 8.8$ | 52.54 |
| | | | GVC | $74.6 \pm 1.0$ | $48.5 \pm 0.1$ | $43.7 \pm 0.4$ | 52.72 |
| | | BBC | ECB+ | $73.6 \pm 1.5$ | $52.5 \pm 1.4$ | $58.3 \pm 1.3$ | 60.26 |
| | | | FCC-T | $71.7 \pm 1.5$ | $53.2 \pm 1.1$ | $60.0 \pm 1.4$ | 60.72 |
| | | | GVC | $73.8 \pm 1.5$ | $52.5 \pm 1.3$ | $58.1 \pm 0.8$ | 60.23 |
| | lemma-$\delta$ | n/a | ECB+ | 73.8 | 40.4 | 46.7 | 50.24 |
| | | | FCC-T | 69.5 | 42.8 | 33.9 | 44.61 |
| | | | GVC | 73.1 | 33.1 | 50.9 | 47.22 |
| $S_{ss}$ | Cattan et al. | ABC | ABC | $75.0 \pm 0.5$ | $46.8 \pm 2.1$ | $58.8 \pm 2.6$ | 58.02 |
| | | BBC | BBC | $66.4 \pm 3.1$ | $51.2 \pm 1.3$ | $60.7 \pm 0.3$ | 58.75 |
| | lemma | n/a | n/a | 61.9 | 42.9 | 33.8 | 43.45 |

Table 15: CoNLL F1 scores

determines optimal labels on entire token sequences instead of scoring individual spans as done by Cattan et al.. While the absolute numbers in the $S_{ss}$ scenario are decent given the circumstances, the differences in mention span definition between HyperCoref and ECB+ evidently are too significant to benefit from HyperCoref for event mention detection.

| Scen. | System | Train | Dev | ECB+ | | | FCC-T | | | GVC | | | H. Mean |
|---|---|---|---|---|---|---|---|---|---|---|---|---|---|
| | | | | P | R | F1 | P | R | F1 | P | R | F1 | F1 |
| $S_{gg}$ | Cattan et al. | ECB+ | ECB+ | $71.4 \pm 0.8$ | $65.2 \pm 1.9$ | $68.2 \pm 1.3$ | $34.6 \pm 1.6$ | $47.1 \pm 2.4$ | $39.8 \pm 0.5$ | $40.0 \pm 8.9$ | $52.9 \pm 7.0$ | $44.6 \pm 4.3$ | 48.22 |
| | | with ABC | ECB+ | $68.3 \pm 2.9$ | $59.0 \pm 3.6$ | $63.3 \pm 3.2$ | $34.8 \pm 1.2$ | $31.9 \pm 7.9$ | $33.0 \pm 4.4$ | $44.9 \pm 6.2$ | $46.6 \pm 8.3$ | $45.0 \pm 1.3$ | 43.91 |
| | | with BBC | ECB+ | $69.6 \pm 1.6$ | $58.3 \pm 0.8$ | $63.4 \pm 0.9$ | $36.3 \pm 0.8$ | $35.6 \pm 4.5$ | $35.8 \pm 2.6$ | $38.5 \pm 2.6$ | $51.8 \pm 1.6$ | $44.1 \pm 1.1$ | 45.19 |
| | | FCC-T | FCC-T | $50.7 \pm 10.2$ | $45.8 \pm 1.3$ | $47.7 \pm 4.3$ | $35.8 \pm 4.3$ | $32.6 \pm 8.0$ | $33.9 \pm 5.8$ | $46.0 \pm 1.6$ | $27.0 \pm 7.3$ | $33.7 \pm 5.9$ | 37.44 |
| | | with ABC | FCC-T | $53.7 \pm 4.9$ | $44.1 \pm 0.9$ | $48.4 \pm 2.5$ | $40.9 \pm 2.6$ | $35.3 \pm 7.3$ | $37.8 \pm 5.3$ | $34.4 \pm 1.5$ | $15.2 \pm 2.1$ | $21.1 \pm 2.3$ | 31.74 |
| | | with BBC | FCC-T | $52.0 \pm 5.8$ | $48.6 \pm 6.9$ | $49.9 \pm 3.5$ | $36.0 \pm 4.2$ | $34.5 \pm 5.6$ | $35.2 \pm 4.9$ | $37.8 \pm 15.0$ | $28.6 \pm 20.7$ | $31.6 \pm 18.3$ | 37.46 |
| | | GVC | GVC | $54.4 \pm 5.7$ | $50.8 \pm 1.3$ | $52.5 \pm 3.0$ | $29.5 \pm 4.1$ | $44.1 \pm 9.2$ | $35.0 \pm 4.5$ | $40.7 \pm 4.4$ | $22.8 \pm 3.5$ | $29.2 \pm 4.0$ | 36.65 |
| | | with ABC | GVC | $50.2 \pm 4.4$ | $51.0 \pm 0.6$ | $50.5 \pm 2.5$ | $31.3 \pm 1.4$ | $48.3 \pm 1.9$ | $37.9 \pm 0.5$ | $50.1 \pm 2.8$ | $27.0 \pm 1.5$ | $35.1 \pm 1.9$ | 40.17 |
| | | with BBC | GVC | $52.1 \pm 10.3$ | $55.2 \pm 1.9$ | $53.1 \pm 4.3$ | $32.0 \pm 1.7$ | $40.1 \pm 12.4$ | $34.9 \pm 4.3$ | $43.8 \pm 4.0$ | $23.1 \pm 6.1$ | $30.1 \pm 6.2$ | 37.17 |
| $S_{sg}$ | Cattan et al. | ABC | ECB+ | $65.2 \pm 3.2$ | $58.9 \pm 2.7$ | $61.8 \pm 0.1$ | $35.0 \pm 2.8$ | $47.1 \pm 1.4$ | $40.1 \pm 1.4$ | $54.4 \pm 4.7$ | $39.2 \pm 7.7$ | $45.0 \pm 3.5$ | 47.36 |
| | | | FCC-T | $70.0 \pm 1.9$ | $55.0 \pm 2.5$ | $61.6 \pm 2.2$ | $38.1 \pm 0.7$ | $39.4 \pm 3.9$ | $38.6 \pm 1.7$ | $43.2 \pm 12.8$ | $20.0 \pm 7.7$ | $27.3 \pm 9.6$ | 38.09 |
| | | | GVC | $70.9 \pm 0.6$ | $55.1 \pm 1.0$ | $62.0 \pm 0.7$ | $37.1 \pm 0.7$ | $40.4 \pm 4.4$ | $38.5 \pm 1.6$ | $44.6 \pm 1.0$ | $19.4 \pm 0.7$ | $27.1 \pm 0.5$ | 37.97 |
| | | BBC | ECB+ | $60.3 \pm 3.5$ | $58.2 \pm 1.6$ | $59.2 \pm 1.8$ | $39.3 \pm 1.4$ | $39.1 \pm 7.6$ | $38.8 \pm 2.9$ | $48.9 \pm 0.9$ | $43.3 \pm 3.0$ | $45.9 \pm 2.1$ | 46.55 |
| | | | FCC-T | $55.6 \pm 3.0$ | $59.2 \pm 2.4$ | $57.3 \pm 2.0$ | $37.1 \pm 1.9$ | $45.4 \pm 7.6$ | $40.5 \pm 2.5$ | $48.1 \pm 0.2$ | $48.2 \pm 3.7$ | $48.1 \pm 1.7$ | 47.67 |
| | | | GVC | $61.1 \pm 3.4$ | $57.6 \pm 0.7$ | $59.3 \pm 1.9$ | $39.2 \pm 1.6$ | $38.7 \pm 6.8$ | $38.6 \pm 2.5$ | $49.2 \pm 1.0$ | $42.8 \pm 1.7$ | $45.7 \pm 1.4$ | 46.40 |
| | lemma-$\delta$ | n/a | ECB+ | 68.6 | 53.5 | 60.2 | 36.0 | 15.7 | 21.8 | 30.0 | 28.3 | 29.1 | 30.98 |
| | | | FCC-T | 56.4 | 50.4 | 53.2 | 38.2 | 19.8 | 26.0 | 8.8 | 29.5 | 13.5 | 22.84 |
| | | | GVC | 70.5 | 53.1 | 60.6 | 34.0 | 7.1 | 11.8 | 48.8 | 27.7 | 35.3 | 23.15 |
| $S_{ss}$ | Cattan et al. | ABC | ABC | $64.0 \pm 3.6$ | $59.4 \pm 2.7$ | $61.5 \pm 0.3$ | $33.8 \pm 3.3$ | $48.1 \pm 1.9$ | $39.6 \pm 1.6$ | $52.8 \pm 3.9$ | $41.5 \pm 7.9$ | $45.9 \pm 3.8$ | 47.39 |
| | | BBC | BBC | $46.1 \pm 4.3$ | $61.3 \pm 0.8$ | $52.5 \pm 2.9$ | $29.4 \pm 4.7$ | $50.8 \pm 4.1$ | $37.1 \pm 4.1$ | $43.9 \pm 1.3$ | $55.7 \pm 2.4$ | $49.1 \pm 0.2$ | 45.20 |
| | lemma | n/a | n/a | 42.8 | 43.5 | 43.1 | 38.4 | 19.9 | 26.2 | 8.8 | 29.7 | 13.6 | 22.24 |

Table 16: LEA scores

| Scen. | System | Train | Dev | P | R | F1 |
|---|---|---|---|---|---|---|
| $S_{gg}$ | Cattan et al. | ECB+ | ECB+ | **48.8** | **72.8** | 58.5 |
| | | w/ ABC | ECB+ | 45.2 | 67.4 | 54.1 |
| | | w/ BBC | ECB+ | 47.1 | 70.2 | 56.4 |
| | Reimers [2018] | ECB+ | ECB+ | n/a | n/a | **79.5** |
| $S_{sg}$ | Cattan et al. | ABC | ECB+ | 37.1 | 55.3 | 44.4 |
| | | BBC | ECB+ | **42.4** | **63.3** | **50.8** |
| $S_{ss}$ | Cattan et al. | ABC | ABC | 37.9 | 56.6 | 45.4 |
| | | BBC | BBC | **42.9** | **63.9** | **51.3** |

Table 17: Event mention detection performance on ECB+. We report mention P/R/F1 on one meta-document of the entire test split. Reimers [2018] is the mean of 25 independent trials.

| Scenario | Train | Dev | Seed | Best Dev... | | |
|---|---|---|---|---|---|---|
| | | | | Epoch | $\tau$ | LEA F1 |
| $S_{gg}$ | ECB+ | ECB+ | 0 | 4 | 0.70 | 59.5 |
| | | | 1 | 16 | 0.70 | 62.3 |
| | | | 2 | 17 | 0.70 | 61.6 |
| | with ABC | ECB+ | 0 | 0 | 0.70 | 55.7 |
| | | | 1 | 2 | 0.65 | 61.0 |
| | | | 2 | 12 | 0.70 | 59.9 |
| | with BBC | ECB+ | 0 | 1 | 0.70 | 58.4 |
| | | | 1 | 10 | 0.70 | 60.4 |
| | | | 2 | 9 | 0.50 | 61.3 |
| | FCC-T | FCC-T | 0 | 7 | 0.70 | 35.1 |
| | | | 1 | 4 | 0.70 | 29.6 |
| | | | 2 | 12 | 0.70 | 29.8 |
| | with ABC | FCC-T | 0 | 23 | 0.70 | 31.3 |
| | | | 1 | 44 | 0.70 | 41.1 |
| | | | 2 | 14 | 0.70 | 33.2 |
| | with BBC | FCC-T | 0 | 2 | 0.65 | 33.1 |
| | | | 1 | 14 | 0.65 | 33.1 |
| | | | 2 | 22 | 0.70 | 40.0 |
| | GVC | GVC | 0 | 3 | 0.55 | 16.2 |
| | | | 1 | 5 | 0.60 | 16.3 |
| | | | 2 | 11 | 0.70 | 16.4 |
| | with ABC | GVC | 0 | 13 | 0.60 | 17.1 |
| | | | 1 | 16 | 0.60 | 17.3 |
| | | | 2 | 11 | 0.60 | 16.3 |
| | with BBC | GVC | 0 | 9 | 0.65 | 16.8 |
| | | | 1 | 16 | 0.70 | 17.7 |
| | | | 2 | 6 | 0.60 | 15.5 |
| $S_{sg}$ | ABC | ECB+ | 0 | 9 | 0.70 | 57.0 |
| | | | 1 | 13 | 0.65 | 57.2 |
| | | | 2 | 21 | 0.70 | 59.6 |
| | | FCC-T | 0 | 9 | 0.65 | 35.8 |
| | | | 1 | 13 | 0.50 | 35.0 |
| | | | 2 | 21 | 0.50 | 36.3 |
| | | GVC | 0 | 9 | 0.55 | 16.5 |
| | | | 1 | 13 | 0.50 | 16.8 |
| | | | 2 | 21 | 0.55 | 12.5 |
| | BBC | ECB+ | 0 | 30 | 0.50 | 56.2 |
| | | | 1 | 30 | 0.55 | 57.4 |
| | | | 2 | 30 | 0.50 | 56.9 |
| | | FCC-T | 0 | 30 | 0.65 | 37.7 |
| | | | 1 | 30 | 0.55 | 39.6 |
| | | | 2 | 30 | 0.55 | 36.8 |
| | | GVC | 0 | 30 | 0.50 | 13.6 |
| | | | 1 | 30 | 0.50 | 13.6 |
| | | | 2 | 30 | 0.50 | 13.2 |
| $S_{ss}$ | ABC | ABC | 0 | 9 | 0.70 | 10.3 |
| | | | 1 | 13 | 0.70 | 13.6 |
| | | | 2 | 21 | 0.70 | 13.8 |
| | BBC | BBC | 1 | 30 | 0.70 | 24.8 |
| | | | 2 | 30 | 0.70 | 25.4 |
| | | | 0 | 30 | 0.70 | 27.1 |

Table 18: Best training epoch, clustering threshold $\tau$ and validation performance per model and independent trial. The best epochs for scenario $S_{sg}$ are taken from $S_{ss}$ and are marked in gray.

