# OpenReview forum: "Event Coreference Data (Almost) for Free: Mining Hyperlinks from Online News"
_AKBC.ws/2021/Conference — AKBC 2021_

### Official Review · Reviewer_bxBV · 2021-07-22
**An interesting semi-automatic approach to producing training data for CDCR that complements other SOTA developments well.**

**Rating:** 7
**Confidence:** 4

**Review:**

This paper proposes an information extraction approach to collecting evaluation and training data for CDCE, relying on the fact that authors of online news
often use hyperlinks to reference significant events when writing online articles. The authors argue that hyperlinks in common for given mentions can be used to form CDCE mention clusters/topics.
The proposed approach tackles the difficulty and cost of producing large-scale CDCE resources, particularly in a multilingual setting.
The authors propose an extraction pipeline to build the HyperCoref corpus from the common crawl (40 sources, 2M mentions) and then proceed to evaluate the quality of the data by evaluating state-of-the-art CDCR systems as compared to performance with currently available gold data, both in-domain and out-of-domain.

The evaluation pipeline appears robust, with precautions taken to remove non event-related hyperlinks (same domain, most frequent prefixes only, no links to high-degree pages).
The authors preform a comprehensive evaluation that shows that system performance with their silver corpus is competitive with using manually curated datasets (at equal size). The qualitative analysis is welcome and offers adequate insights into the intrinsic quality of the data.




*Strengths*:
- Well written and easy to follow
- Elegant yet simple approach
- Robust pipeline that's easy to transpose to arbitrary source sites.

*Shortcomings*:
- Since multilinguality is one of the key arguments to generate silver-data, some limited evaluation and analysis would have been welcome (it's only evoked that experiments were performed and that the approach also works).


*Other questions / comments*:

Couldn't exceptionsbe added for links to trusted domains (for example to wikipedia pages, which are often referenced in articles, even tough they aren't always event-oriented). ?

---

> ### Author Response · Authors · 2021-07-30
> **Response**
>
> Thank you for your comments.
>
> * We agree that deeper analysis of the language independence of the proposed approach would have strengthened the submission. However, since proper multilinguality research would have required checking multiple language families, sourcing appropriate gold data for comparison (which is hard to come by) and researchers capable of these languages, we did not pursue this idea further. We will however release the implementation of the entire data pipeline, hence researchers willing to address multilinguality may do so in the future.
>
> * Regarding trusted domains: We had tried this during development, but found that it predominantly adds noise. From what we remember anecdotally, links to Wikipedia mostly referenced people (actors, politicians, etc.), countries/regions or other things people would consider looking up in an encyclopedia, but rarely eventive content.
> We also tried permitting links between trusted news outlets (such as including hyperlinks to bbc.com in articles published by ABC News). This worked sometimes, but in the vast majority of cases it only led to citation-like hyperlinks such as "as per the BBC", "as reported by the BBC", or just "[BBC]". Our implementation still is capable of whitelisting arbitrary domains, so researchers with a good idea on how to filter these cases quickly and reliable may experiment freely.

---

### Official Review · Reviewer_cWi5 · 2021-07-23
**Solid paper on x-doc coref corpus from newswire w/ hyperlinks**

**Rating:** 7
**Confidence:** 4

**Review:**


This isn't going to move the field forward by a huge amount, but its a bit of solid work and above the bar in my estimation.

I was forced due to time constraints to give this a very quick review, a deeper one (due shortly) may answer some of these questions and concerns:

1) There has a been a lot of work on using hyperlinks for entity detection, and event detaction in twitter, the claim to novelty is based on the scarcity of that in news corpora.  This needs to be checked better (on me), I didn't see enough of that in the related work section.

2) I would have liked to see more direct comparison to WEC.  The papers says: "These corpus differences have a direct impact on results, with our broad-coverage ABC Sss model outscoring the identical RoBERTa-based architecture trained by Eirew et al. [2021] on WEC by 5.9% CoNLL F1 when tested on ECB+ (75.04 vs. 69.10 F1)."

Is this direct comparison valid? You don't really say how you're using CoNLL F1, which in my understanding is based on OntoNotes, but that doesn't seem to agree with the text.  Please add more explanation of how you used this metric.  The moschitti CoNll paper says the official metric is "a weighted average of three metrics: MUC, B-CUBED, and CEAF" - is that what you did?

Anyway, I suspect direct comparison of WEC and Hypercoref is apples-oranges unless you had the same test set, so I would love to see a system trained on HC perform on WEC, and vice-versa, it would be more conclusive....

(confidence reflects high expertise but very quick read of the submission)

---

> ### Author Response · Authors · 2021-07-30
> **Response**
>
> Thank you for your comments.
>
> ### Concern 1
> We were not aware of additional related work in this area. If you find the time, we would highly appreciate if you could point us to the additional related work you mentioned.
>
> ### Concern 2
> Please excuse the dense sentence, we meant to save space there. We did the following:
> 1. We trained the RoBERTA-based model of Arie Cattan et al. 2020 entirely on silver data from HyperCoref (specifically the ABC subcorpus).
> 2. We use the Python-based coreference scorer implementation from Moosavi et al. (see Appendix A.4.1) which produces the same results as the official Perl-based scorer [1] but additionally reports the LEA coreference metric.
> 3. Prior to running the coreference system, we simplify the task by pre-partitioning all documents according to Barhom et al. 2019 (see Appendix A.3 in the third-to-last paragraph).
> 4. To evaluate cross-document coreference, we merge all documents into one meta-document and run the scorer on this document. This is the quasi-standard procedure since at least Barhom et al. 2019's paper.
> 5. We test on the official ECB+ test split.
>
> Eirew et al. 2021 did the following:
> 1. They train a RoBERTA-based model on silver data from WEC. Based on their model description and their code on Github, the Eirew et al. 2021 is equivalent to the Cattan et al. 2020 model (also note that Arie Cattan is a co-author of the Eirew et al. 2021 paper).
> 2. They use the official Perl-based scorer [1].
> 3. and 4. and 5. are reported in their paper and match our setup.
>
> So, to the best of our knowledge, the comparison is valid.
> Please also refer to our response to "Reviewer 1ase" as to why WEC is discussed rather briefly in our work.
>
> [1] https://github.com/conll/reference-coreference-scorers

---

### Official Review · Reviewer_QASE · 2021-07-23

**Rating:** 9
**Confidence:** 3

**Review:**


This paper collects and releases a substantial new resource for the challenging task of cross-document event coreference, of "silver" coreference clusters automatically constructed from hyperlinks in web news articles.  It is substantially larger than previous human-annoated cross-doc event coref datasets, which is a task that is extremely challenging to annotate.  The authors also demonstrate that training a model on this silver data yields a high-performing model as evaluated on a previous human-annotated dataset.

I very strongly believe there are important event KB extraction applications which need more research attention, which is the main justification for event coref research, since it could be useful to help them - for example, the Gun Violence Corpus, drawn from efforts to document gun violence, which is used as part of the evaluation - and data availability for event coref is a major issue.  It still seems like a long way to go to practical usability, but that's to be expected on a challenging task.  One thing I'm less sure of is whether cross-doc event coref as mention clustering, in the way it's conceptuatlized from ECB through this work, has the potential to actually be useful for event KB extraction, as opposed to doing something more direct.  Cluster-style event coref has all the many conceptual limitations spelled out in the Hovy et al. "Events are not Simple" paper (in the references).  Still, it's not obviously a wrong approach to the problem, and certainly worth exploring.  Note that this paper briefly presents different justifications for event coref - QA and fact-checking - but without many details on how event coref could be applied.

The paper also has a good description of the process used to create the resource, which could be helpful for other efforts to exploit web news and hyperlink structure for other semantic tasks; it seems the choices and heuristics used took a fair amount of development themselves, and a successful previous example could help future guide efforts both for this task, or other fine-grained cross-document tasks.

The performance story about how useful their silver data is, is somewhat complicated in my opinion - it is certainly decent by itself, but if you have access to gold data, it does not always uniformly enhance the model.  Perhaps there is room for future modeling advances to better account for the noisiness in silver data input.

It would be helpful to discuss the motivation for only using ABC and BBC sources for news data.  Was there something about their hyperlink writing style that made them more useful?  Or was it due to size?  Was there a need to customize preprocessing for each website?

minor
 - pg 5: "fine-granular" => "fine-grained"

---

> ### Author Response · Authors · 2021-07-30
> **Response**
>
> Thank you for your comments.
>
> * Regarding fact checking, one possibility of applying CDCR we imagine would be to find additional evidence in a large corpus on the level of statements (not entire documents). QA systems could benefit from a network of interrelated statements (connected by the event coreference relation) to source additional material for answers.
> As you probably had suspected, these would be retrieval-style applications rather than clustering applications.
> Apart from the conceptual limitations of clustering you already mentioned, the established evaluation methodology for CDCR makes it difficult to distinguish cross-document from within-document performance (see Upadhyay et al. 2016 in the references). From a technical standpoint, existing CDCR systems could be adapted relatively easily to a retrieval setup, since they learn either pair similarity or vectorized mention representations from which a ranking could be derived.
> Still, since the focus of this particular work was the development of the dataset, we stuck with the established way of evaluating CDCR through clustering for comparability.
>
> * Regarding the performance and choice of news outlets:
> We realize that we have only scratched the surface with respect to modeling experiments in this work. The purpose behind the experiments conducted mainly was to assess whether hyperlink data can be viable training data for CDCR at a fundamental level.
> We picked ABC and BBC since they are reputable news sources, have different composition of domains (see Table 1b), and are mid-to-large in size (see Appendix Table 4), i.e. a representative sample of the whole dataset. We would have liked to train on more datasets, but already faced scalability problems with the SOTA system implementation on this small subset (see our comment on page 7 under "Data Preparation" and the second paragraph in Appendix A.3, page 25). We are considering future work on the modeling side to rectify this.
>
> * Regarding customized preprocessing per website:
> While developing the handwritten rule set (step 5d on page 4), we gave the extracted data from each news outlet a cursory look to make sure we did not miss major sources of noise. Apart from this all pipeline settings are website-agnostic.

---

### Official Review · Reviewer_1ase · 2021-07-23
**strong paper introducing new large-scale CDCR resource, strong results on existing gold datasets**

**Rating:** 7
**Confidence:** 3

**Review:**

The paper collects a new ‘silver’ dataset HyperCoref for cross document coreference resolution task, constructing by using hyperlinks in online news articles. The paper shows that the dataset is considerably larger () than existing human-annotated datasets and has broader coverage in terms of event types. Experiments on three existing human-annotated datasets (ECB+, FCC-T, GVC) show that models trained on the silver-dataset perform comparable to those trained on gold train data.


Strengths:
1. The dataset collected is significantly larger than existing resources and will be useful for future research in this area.
2. The collection pipeline is well-described. The filtering and hand-written rules appear to be reasonable and can potentially be extended to other sources/languages.
3. The analysis of the silver data, and its comparison to existing datasets, provides useful insights about the data distribution.
4. Models trained on the collected ‘silver’ data show impressive results on gold test sets.


Weaknesses:

1. Direct comparison with WEC is missing. I am not fully convinced by the argument that the events in WEC are inherently limited. Have you tested how well models trained on HyperCoref perform on the WEC test set?

---

> ### Author Response · Authors · 2021-07-30
> **Response**
>
> Thank you for your comments.
>
> The reason why WEC is discussed only briefly is that at the time of WEC's announcement (end of April), experimental work for this paper was nearly complete. We therefore did not test HyperCoref models on WEC, though we agree that it would be an interesting scenario for the future.
>
> Regarding our comments on the limitations of events in WEC, we give a deeper explanation here:
> Events in WEC stem from Wikipedia articles on real-world events. The Wikipedia community has established a set of guidelines for judging whether a real-world event is notable enough to warrant an article [1]. These guidelines require an event to have lasting effects and sufficient geographical scope. Writing articles for "routine events" (specifically mentioned are "most crimes, accidents, deaths, celebrity or political news") is discouraged. Conversely, crimes, accidents, celebrity or political news are the predominant types of events present in the ECB+, FCC-T and GVC corpora.
> Due to this mismatch, we see only limited potential for (pre-)training CDCR systems on WEC if the goal is to apply those systems on ECB+/FCC-T/GVC. There may be other applications (possibly involving question answering) where the encyclopedic origins of WEC could be beneficial though.
>
> [1] https://en.wikipedia.org/wiki/Wikipedia:Notability_(events)

---

### Decision · Program_Chairs · 2021-08-17

**Decision:**

Accept

**Comment:**

This paper presents a silver-standard event coreference corpus based on mining hyperlinks: if two hyperlinked event phrases point to the same article, the hyperlinked phrases are interpreted to refer to the event described in that article. The results show that a model trained on this data achieves decent out-of-domain performance on ECB+, FCC-T, and GVC, although this performance isn't complemented that strongly by the presence of a small amount of gold data. The reviewers generally agreed that this was solid work, presenting a sensible data collection process yielding a sizable dataset that should be helpful to researchers in this area. One of the reviewers raised some concerns about sticking with the clustering-based and the potential lack of utility of the silver-standard data, but concedes that perhaps better modeling or training could address the latter point. Other concerns were raised about similarities to the WEC corpus; however, we are reasonably convinced by the counterarguments about the skewed distribution of what gets linked in Wikipedia and the fact that WEC was published so close to the submission deadline for this work.